**communications** engineering

# Integrating fixed and mobile coherent optical access networks for unified broadband services
Qi Wu [1] ✉, Zixian Wei[1], Xiaoying Zhang[1,2], Haiqiang Wei[1], Jiahao Huo [2], Xinran Huang[3], Tonghui Ji[4], Zhaopeng Xu [4], Chao Lu[1], Alan Pak Tao Lau[1] & Kangping Zhong [1] ✉

The integration of fixed and mobile optical access networks can enhance resource utilization, reduce network complexity, and lower deployment costs. Here, we report a field trial demonstrating the integration of hardware-efficient coherent optical transmission and high-fidelity analog waveform delivery over a single optical carrier for converged fixed-mobile networks, enabled by the proposed amplitude-phase layered modulation. This approach not only simplifies the digital coherent reception for fixed optical access, but also enhances analog transmission by enabling signal-to-noise ratio adaptation for 6G mobile front-haul. We experimentally validate the proposed scheme in a 109-km field-deployed fiber link in Hong Kong. A 128 Gb/s phase-insensitive digital coherent transmission is successfully integrated with 20-GHz aggregation bandwidth wireless signals, supporting single-wavelength 1.2 Tb/s common public radio interface equivalent data rate and 256-ary quadrature-amplitude-modulation format. Using this scheme, the fixed optical access achieves a power budget exceeding 38.6 dB, while analog mobile front-haul signals reach signal-to-noise ratios above 47.2 dB and support modulation formats up to 16384-ary quadrature-amplitude-modulation. These results highlight the potential of this solution to unlock the capacity of existing and future fiber-optic infrastructures by converging fixed access, mobile access, and metro networks within a unified framework that enhances scalability and simplifies network deployment.

Optical networks play a pivotal role in meeting the ever-growing bandwidth demands of modern society. They span core, metro, data center, and access networks. Within the access segment, fixed access systems—such as passive optical networks (PONs)[1–4]—and mobile access systems, particularly mobile fronthaul[5–8], are of critical importance, as they form the foundation of next-generation communication systems. With the advent of the artificial intelligence era and the rapid proliferation of data-driven applications, network traffic is surging at an unprecedented rate. As a result, optical access networks are facing persistent capacity expansion challenges, requiring scalable, efficient, and future-proof solutions to accommodate the rapidly evolving connectivity demands.

In the realm of fixed optical access, PON technology has evolved significantly. Starting from early generations such as Gigabit-capable PON and 10-Gigabit Symmetric PON (XGS-PON), the industry has now entered the stage of field trials and early deployments of 50G-PON[9], guided by the

International Telecommunication Union - Telecommunication Standardization Sector (ITU-T) standards[10], which promise substantially higher bandwidth and an improved user experience, with large-scale roll-outs expected in the coming years. Looking ahead, next-generation PON systems are expected to adopt coherent detection techniques, enabling 100G and even 200G coherent-PON solutions[11,12]. Coherent detection can exploit multiple modulation dimensions—including phase and polarization—to dramatically increase spectral efficiency[13,14]. Moreover, it offers robust compensation for optical impairments such as chromatic dispersion (CD) and polarization mode dispersion[15,16], while also improving receiver sensitivity due to the beating gain provided by a local oscillator (LO). Although coherent systems, compared with direct detection, require frequency offset estimation (FOE) and carrier phase recovery (CPR) in receiver-side digital signal processing (DSP)—resulting in higher power consumption and latency—their advantages still position coherent PON as a strong candidate for future high-capacity, high-sensitivity access networks, pushing the limits

[1]Photonics Research Institute, Department of Electrical and Electronic Engineering, The Hong Kong Polytechnic University, Hong Kong, China. [2]School of Computer and Communication Engineering, University of Science and Technology Beijing, Beijing, China. [3]China Telecom Research Institute, Shanghai, China. [4]Pengcheng Laboratory, Shenzhen, China. ✉e-mail: polyu-qi.wu@polyu.edu.hk; kangping.zhong@polyu.edu.hk

of performance while preserving the cost-effectiveness essential for large-scale deployment.

On the mobile access side, the concept of centralized radio access network (C-RAN) has attracted considerable traction, where fronthaul technologies such as the common public radio interface (CPRI)[17] and its evolved successor, the enhanced CPRI (eCPRI)[18]—an industry-standard specification that supports efficient packet-based transport over Ethernet networks—serve as key enablers. These digital interfaces connect remote radio units (RRUs) to centralized baseband units over optical fiber, allowing for efficient coordination and resource pooling. However, each gigahertz of wireless bandwidth typically requires approximately 61.4 Gb/s of fronthaul capacity when using digitized CPRI in representative mobile network configurations, resulting in very low wireless-to-optical bandwidth efficiency[7,19,20]. Such inefficiency makes it impractical to accommodate the terabit-per-second rates envisioned for 6G networks[21]. To address this limitation, analog radio-over-fiber (A-RoF) has emerged as a promising fronthaul solution[22–28]. Unlike digital interfaces, A-RoF maps the original radio frequency (RF) signals directly onto optical carriers, ideally occupying the same bandwidth as the wireless signal. This approach offers inherent advantages, including reduced system complexity, low latency, and high spectral efficiency[29]. However, conventional A-RoF is limited by a signal-to-noise ratio (SNR) ceiling of approximately 25 dB[30], imposed by system impairments such as transceiver nonlinearities and electrical noise, which restricts its support to modulation formats no higher than 64-ary quadrature-amplitude-modulation (QAM)[19]. Furthermore, the link SNR is physically fixed by the analog front end, leaving little flexibility to adjust signal fidelity. Given that current and future wireless standards demand formats up to 1024-QAM and beyond[31], the performance of traditional A-RoF remains insufficient. Consequently, the choice between digital and analog fronthaul architectures remains an open question[20,32], critically shaping the evolution of mobile access networks to support the demanding requirements of future 6G systems.

In the past, wireless and optical access networks were built as independent systems. This parallel deployment is now proving inefficient, as it duplicates resources and struggles to meet surging bandwidth demands. To address these challenges, operators are moving toward unified architectures that combine fixed and mobile access[33–35]. This convergence primarily facilitates the sharing of existing infrastructure such as optical line terminals (OLTs), resulting in substantial reductions in both capital and operational expenditures while facilitating streamlined service delivery[36,37]. Most recently, several integrated network architectures[38–50] have been proposed to simultaneously deliver digital bit streams and analog wireless signals over shared optical links. These approaches exploit orthogonal transmission resources—including wavelength[38,40,49], polarization[45], and frequency[44,46,48,50] to multiplex different signal types. Such strategies open promising avenues for the realization of future-proof, multi-service access networks that can accommodate both high-speed broadband and low-latency wireless fronthaul services[51]. However, these solutions often rely on complex and heterogeneous transceiver architectures to multiplex and de-multiplex digital and analog signals, limiting their practicality for unified access network deployment. More importantly, current approaches are constrained by the inherent SNR ceiling of optical links, which prevents the simultaneous delivery of ultra-high-fidelity wireless signals beyond 256-QAM and CPRI-equivalent data rates exceeding the terabit-per-second level. Furthermore, demonstrating the true feasibility of fixed-mobile access network integration necessitates proof-of-concept trials over field-deployed fiber infrastructure, as real-world conditions introduce diverse impairments—such as temperature fluctuations and mechanical vibrations—that cannot be fully replicated in laboratory experiments[52,53].

In this work, we experimentally demonstrate an integrated fixed-mobile optical access network architecture through laboratory experiments and transmission over field-deployed fibers, enabled by the proposed amplitude-phase layered modulation. This scheme allows for simplified coherent DSP by eliminating the need for FOE and CPR, while simultaneously supporting high-fidelity analog signal delivery using high-order

modulation formats. Specifically, digital signals employing unipolar pulse amplitude modulation (PAM) are mapped onto the optical amplitude, whereas the optical phase carries wireless orthogonal frequency division multiplexing (OFDM) signals. The amplitude-phase layered modulation benefits both digital and analog transmissions. For digital signals, encoding the information in the optical intensity eliminates the need for FOE and CPR at the receiver side, thereby reducing computational complexity, lowering power consumption, and minimizing processing latency. For analog signals, the presented phase modulation (PM) enhances SNR through spectral expansion by increasing the modulation index[54], enabling the transmission of high-fidelity analog OFDM signals with higher-order modulation formats beyond 256-QAM. This work extends our post-deadline paper presented at ECOC 2025[55]. It introduces a comprehensive fixed-mobile converged network architecture, provides a detailed elucidation of the underlying principles, presents substantial additional experimental results, and includes an in-depth discussion encompassing thorough performance comparisons, compatibility with existing systems, low-cost implementation alternatives, and considerations for upstream transmission. Here, we experimentally demonstrate the proposed scheme over both 20-km laboratory fiber and 109-km field-deployed fiber in Hong Kong, achieving successful integration of 128 Gb/s digital bits with dual-polarization aggregated analog bandwidths of 36, 20, 16, 8, and 4 GHz for 64-QAM, 256-QAM, 1024-QAM, 4096-QAM, and 16384-QAM, respectively. For digital transmission, the achieved optical power budget exceeds 38.6 dB, surpassing the ITU-T's highest E2-class requirement of 35 dB[10] for optical network units (ONUs). For analog transmission, the corresponding SNRs range from 23.1 dB for 64-QAM to 47.2 dB for 16384-QAM, supporting CPRI-equivalent data rates of up to 2.4 Tb s$^{-1}$ with 64-QAM and 1.2 Tb s$^{-1}$ with 256-QAM. This dual-benefit integration approach paves the way for the efficient convergence of fixed and mobile access networks, offering a unified infrastructure for future broadband access networks and providing a new idea for the co-delivery of digital and analog signals.

## Methods

### Amplitude-phase layered modulation for fixed-mobile access network integration

Figure 1a illustrates the proposed integrated fixed-mobile access network, which enables the coexistence of PON technologies and C-RAN. This architecture supports a wide range of services and applications, including fiber-to-the-X (FTTx) deployments (X: antenna, building, home, node), massive cellular connectivity for 6G, intelligent transportation systems, and the industrial Internet of Things. To realize this converged network, we propose an amplitude-phase layered modulation scheme, illustrated in Fig. 1b. The digital bit stream and analog waveforms can be transmitted simultaneously over a single optical carrier frequency, supporting a unified deployment for standard polarization-diverse coherent receivers at both ONUs and RRUs. For clarity, only one polarization is illustrated. Nonetheless, this arrangement naturally enables the incorporation of polarization-division multiplexing[15], effectively doubling capacity and improving spectral efficiency. Furthermore, this scheme eliminates the need for devices to multiplex and de-multiplex digital and analog signals, enabling faster routing, leveraging existing multi-user access infrastructure, and supporting highly flexible network architectures.

In this approach, the amplitude and phase of the optical carrier are utilized to convey digital and analog information, respectively, thereby achieving dual functionality. Specifically, the optical electric field can be expressed as $E(t) = A(t)e^{j\Phi(t)}$, where $A(t)$ denotes the optical amplitude, typically used to encode digital bit streams (e.g., a PAM signal $S_D(t)$), and $\Phi(t)$ represents the optical phase, which is modulated by analog signals with a flexible modulation index (e.g., wireless OFDM signals $k_p S_A(t)$). This enables phase-insensitive coherent detection for ONUs across various deployment scenarios, such as FTTH, FTTN, and FTTB, while allowing the application of spectrum expansion techniques to overcome channel-limited SNR bottlenecks, thereby ensuring high-fidelity wireless signal

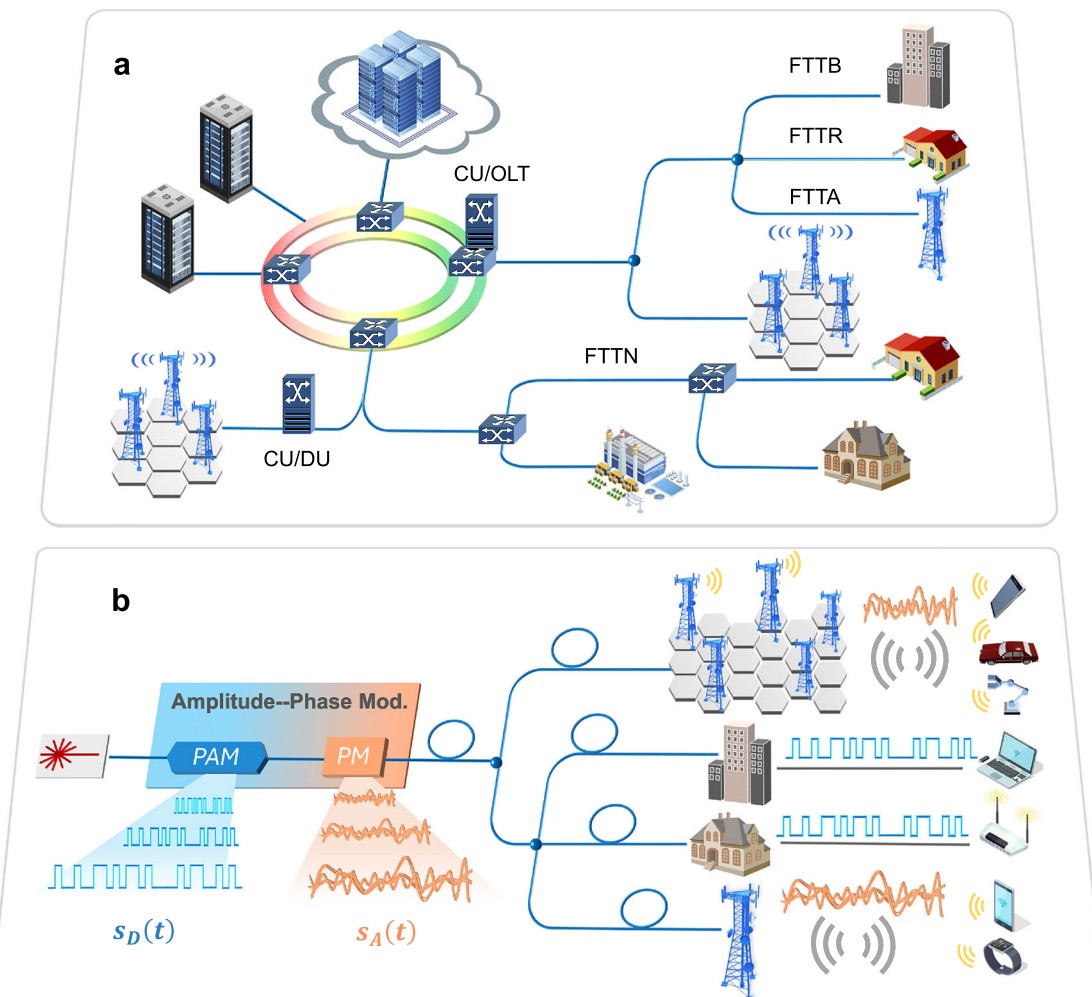

**Fig. 1 | Next-generation integrated fixed-mobile coherent optical access network architecture. a** Schematic of the converged access network integrating multiple ONUs and dense cellular deployments. CU Central Unit, DU Distributed Unit, FTTB Fiber-to-the-Building, FTTA Fiber-to-the-Antenna, FTTH Fiber-to-the-Home, FTTN Fiber-to-the-Node, OLT Optical Line Terminal. **b** Concept of the proposed amplitude-phase layered modulation scheme. PAM Pulse Amplitude Modulation, PM Phase Modulation.

transmission for RRUs. The detailed operating principles of the proposed amplitude-phase layered modulation are described in the following context.

### Unipolar PAM resilient to frequency offset and phase noise

Traditional bipolar PAM encodes bits onto the amplitude of light (e.g., PAM-2 with symbols ± 1), resulting in an optical field of the form $S_D(t)e^{j(2\pi f_c t+\phi(t))}$, where $f_c$ and $\phi(t)$ are the optical carrier frequency and phase of the transmitter laser, respectively. In the case of coherent detection, with a frequency offset $\Delta f(t)$ and phase noise $\Delta\phi(t)$ introduced by the incoherence between the transmitter and receiver lasers, the received optical field can be expressed as

$$E_{rx}(t) = S_D(t)e^{j2\pi\Delta f(t)t+j\Delta\phi(t)}, \tag{1}$$

which forms ring-like constellations, as illustrated in Fig. 2a. In such a scenario, the information encoded in the sign of the digital signal becomes embedded within the phase variations, making reliable detection challenging. Therefore, FOE and CPR must be performed in the DSP to correctly extract the signs of the digital symbols.

To address this issue, we implement a mapping of bits onto unipolar PAM symbols (e.g., two levels 1 and 3) to eliminate phase ambiguity, as

illustrated in Fig. 2b. In this case, even if the lasers introduce frequency offset and phase noise, the received signal still forms distinguishable rings. The amplitude of the signal can then be directly extracted using an absolute-value operation to remove the impact of frequency offset and phase noise. Compared with conventional bipolar PAM, the proposed phase-insensitive unipolar PAM modulation not only simplifies coherent DSP for the digital layer—eliminating the need for FOE and CPR—but also enables simultaneous phase modulation for analog fronthaul signals, thereby overcoming the inherent SNR ceiling in traditional analog radio-over-fiber links.

### Scalable SNR adaptation for high-order QAM via phase modulation

To overcome the inherent SNR limitation of optical fiber links—typically around 25 dB when considering the transceiver electrical noise floor and optical impairments[19,31]—we introduce phase modulation superimposed on phase-insensitive digital signals. The transmitted optical field can be expressed as

$$E_{tx}(t) = S_D(t)\,e^{jk_p S_A(t)}, \tag{2}$$

where $S_D(t)$ represents the unipolar digital signal, $S_A(t)$ denotes the analog waveform to be transmitted, and $k_p$ is the phase modulation index (radians

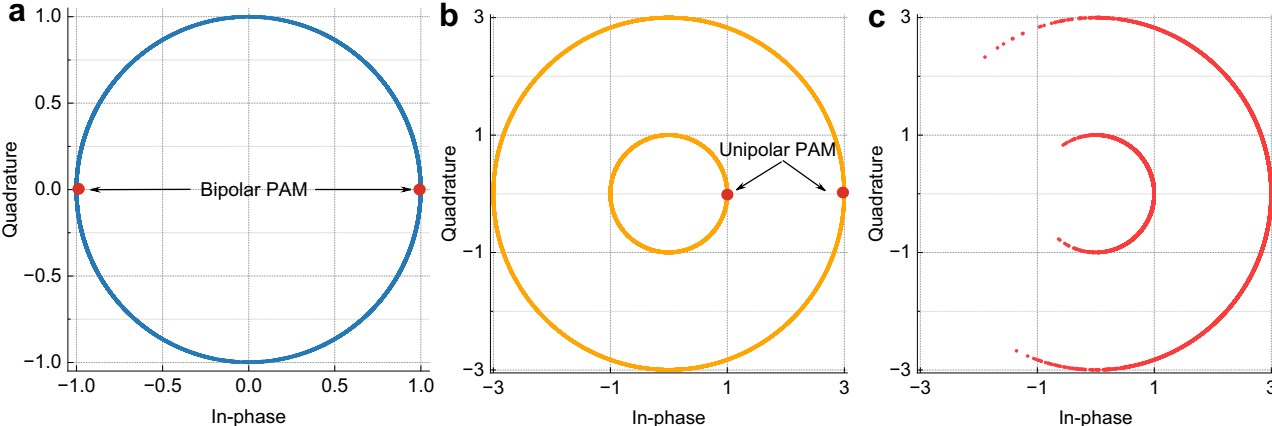

**Fig. 2 | Modulation constellations. a** Bipolar pulse amplitude modulation under frequency offset and phase noise. **b** Phase-insensitive unipolar pulse amplitude modulation under frequency offset and phase noise. **c** Amplitude-phase layered modulation.

per unit amplitude). The constellation of the amplitude-phase layered modulation is shown in Fig. 2c. This scheme enables SNR scaling with an increasing modulation index. According to classical communication theory[54], the SNR in linear units of the recovered analog signal under ideal coherent detection for a PM waveform is given by

$$\text{SNR}_{\text{PM}} = \frac{k_\text{p}^2 \langle S_D^2(t) \rangle \langle S_A^2(t) \rangle}{2N_0 B}, \quad (3)$$

where $\langle S_D^2(t) \rangle$ and $\langle S_A^2(t) \rangle$ denote the mean power of the digital and analog signals, respectively; $B$ is the original bandwidth of $S_A(t)$; and $N_0$ is the noise power spectral density of the channel. The SNR in (3) is expressed in a linear scale. It is evident that $\text{SNR}_{\text{PM}}$ scales quadratically with the modulation index $k_\text{p}$, indicating that doubling $k_\text{p}$ theoretically results in a fourfold SNR increase, corresponding to an improvement of approximately 6.02 dB. In A-RoF systems, phase modulation provides a key advantage: SNR adaptability. In theory, arbitrarily high-order modulation formats can be supported by appropriately selecting the modulation index. Moreover, because the information is encoded in the instantaneous phase rather than the amplitude, phase modulation is inherently more robust against optical power fluctuations and amplitude-related noise sources, such as laser relative intensity noise or the pattern-dependent effects of semiconductor optical amplifiers.

However, this SNR scaling is achieved at the expense of an increased occupied bandwidth, as a higher modulation index introduces additional high-order terms in the Taylor expansion of $e^{jk_\text{p} S_A(t)}$. The resulting signal bandwidth after spectral expansion can be approximated using the well-known Carson's rule[54]:

$$B_T = 2(D+1)B, \quad (4)$$

where $B_T$ is the total occupied bandwidth after phase modulation, $B$ is the baseband bandwidth of the analog signal $S_A(t)$, and $D = \frac{\Delta f}{B}$ is the modulation index or frequency deviation ratio, with $\Delta f = k_\text{p} \max|S_A(t)|$ representing the peak frequency deviation. It should be noted that, within this integration scheme, the bandwidth of the digital signal $S_D(t)$ generally remains higher than the expanded analog bandwidth $B_T$, to fully utilize the channel bandwidth.

Overall, this property of amplitude-phase layered modulation enables SNR scaling through an increased modulation index, thereby supporting high-fidelity analog signal transmission together with higher-order modulation formats in the analog layer. Moreover, the phase-insensitive nature of the digital modulation ensures reliable data recovery even in the presence of substantial phase noise and frequency offset, facilitating the seamless

coexistence of high-SNR analog modulation and power-efficient digital transmission on a shared optical carrier.

## Results
### Experimental setup for converged fixed-mobile optical access networks

Figure 3 illustrates a proof-of-concept demonstration of a short-reach converged PON and mobile fronthaul network based on a standard coherent transceiver architecture. The design employs a dual-polarization IQ transmitter and intradyne coherent receivers, ensuring compatibility with existing coherent optics platforms. Further discussion regarding the hardware configuration, including the implementation based on intensity modulation and direct detection (IM-DD), is provided in the *Discussion* section. Specifically, the transmitter consists of four digital-to-analog converters (DACs, Keysight 8196A) operating at 90 GSa/s with 8-bit vertical resolution and 32-GHz bandwidth. Therefore, the symbol rate of the digital signal is fixed to 64 GBd. On the receiver side, an integrated intradyne coherent receiver chip (Lumentum Class 40) is employed together with a real-time oscilloscope (Keysight DSAX96204Q), providing four analog-to-digital converters with a sampling rate of 80 GSa/s and 33-GHz bandwidth.

The RF and digital signal operations are emulated offline via DSP, as illustrated in insets (i-ii) of Fig. 3. At the transmitter DSP, random digital bits are first generated and mapped into unipolar PAM symbols. Meanwhile, QAM signals with constellation sizes ranging from 256 to 16384 are generated to emulate RF signals after frequency down-conversion. OFDM is employed as a generic method to emulate the aggregation of heterogeneous RF signals[19,20]. The OFDM signal is generated using a 1024-point inverse fast Fourier transform (IFFT). Among the 1024 subcarriers, 900 are employed as effective data-carrying subcarriers, while the remaining subcarriers are zero-padded at the spectrum edges to prevent aliasing. A cyclic prefix of 8 samples is appended to each OFDM symbol to mitigate inter-symbol interference arising from channel dispersion. Note that key parameters, including the IFFT size, number of active subcarriers, and cyclic prefix length, can be adjusted depending on the specific experimental requirements and bandwidth configuration[56]. The analog signal is then phase-modulated with a flexible modulation index ($e^{jk_\text{p} S_A(t)}$). The digital and analog signals are combined to realize the amplitude-phase layered modulation scheme. The composite signal is subsequently up-sampled and shaped by a root-raised-cosine filter with a roll-off factor of 0.01. A -32.82 GHz pilot tone is inserted to facilitate carrier phase recovery[13] at the RRUs. After resampling to match the DAC sampling rate, the generated electrical waveforms are used to drive the DP-IQ modulator (Fujitsu FTM7992HM) with an insertion loss of 14 dB and 3-dB bandwidth of 35 GHz, which is biased at the null point to modulate the continuous-wave output from an external cavity laser (Ovlink TSP-1000) with a linewidth of 100 kHz, centered at 1549.49 nm. The output

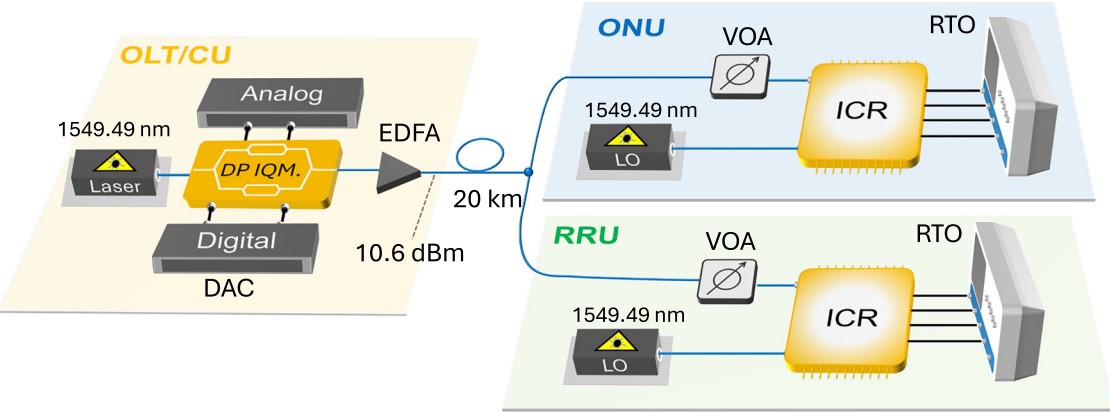

| **(i) Tx DSP (Digital)** | **(ii) Tx DSP (Analog)** | **(iii) Rx DSP (ONU or RRU)** |
|---|---|---|
| Random bits generation | Baseband OFDM generation | Resampling |
| Unipolar PAM modulation | Phase modulation | FOE and CPR (RRU only) |
| Amplitude and phase combination ($S_D(t)e^{jk_p S_A(t)}$) | | 2×2 MIMO Equalization |
| Up-sampling and RRC pulse shaping | | Down-sampling |
| RF pilot tone insertion | | Amplitude & Phase Demodulation |
| Resampling and loading to DACs | | BER & QAM SNR/EVM calculation |

**Fig. 3 | Experimental setup and digital signal processing workflow at the ONU and RRU sides.** CU Central Unit, OLT Optical Line Terminal, DP IQM. Dual-Polarization IQ Modulator, DAC Digital-to-Analog Converter, RRU Remote Radio Unit, ONU Optical Network Unit, VOA Variable Optical Attenuator, EDFA Erbium-Doped Fiber Amplifier, LO Local Oscillator. ICR Intradyne Coherent Receiver, RTO Real-Time Oscilloscope, OFDM Orthogonal Frequency Division Multiplexing, RRC Root-Raised Cosine, PAM Pulse Amplitude Modulation, FOE Frequency Offset Estimation, CPR Carrier Phase Recovery, MIMO Multi-Input Multi-Output.

power of the modulator is −10.4 dBm. Thus, the optical signal is then amplified to 10.6 dBm using an Erbium-doped fiber amplifier (EDFA, Amonics) operating with a gain of 21 dB, before being transmitted over a 20-km single-mode (SMF) fiber link. The launch power of 10.6 dBm is selected through experimental optimization. Higher launch powers would induce fiber nonlinearities—primarily self-phase modulation—leading to degradation in system performance.

At the receiver side, we employ a variable optical attenuator (VOA) to emulate various splitting ratios introduced by the optical splitter in a typical point-to-multipoint access network[3], thereby evaluating the received optical power (ROP) sensitivity and optical power budget. Conventional intradyne coherent detection is employed at both the ONU and RRU, with the local oscillator (LO) wavelength fixed at 1549.49 nm, identical to that of the transmitter. However, due to the inherent frequency drift of the lasers, a frequency offset of approximately 200 MHz exists between the transmitter and the LOs. It leverages the enhanced receiver sensitivity provided by the LO beating gain while fully utilizing the receiver's electrical bandwidth. The acquired waveforms are then resampled to two samples per symbol for subsequent offline DSP. The DSP workflow for the two nodes is largely similar, with the exception that FOE and CPR are omitted at the ONU owing to its phase-insensitive modulation. The DSP flow, illustrated in inset (iii) of Fig. 3, consists of routine coherent-DSP operations, including resampling, pilot-tone-based FOE and CPR (for the RRU only), frame synchronization, 2 × 2 time-domain radius-directed multi-input-multi-output (MIMO) equalization with 15 taps to mitigate inter-symbol interference, and down-sampling. Finally, the digital symbols and analog waveforms are demodulated simply by applying absolute-value and angle operations to the down-sampled symbols. The amplitude-modulated digital signal is evaluated using bit error rate (BER). For the phase-modulated layer carrying analog fronthaul signals, transmission fidelity is assessed via SNR and error vector magnitude (EVM)[19,28,42], where EVM (linear) $\approx 1/\sqrt{\text{SNR}}$.

This aligns with relevant industry standards (e.g., 3GPP TS 38.101-2[56]), which specify EVM as the primary quality metric for wireless signals.

**Performance of integrated digital and analog transmission**
Figure 4a illustrates the digital spectra of a 1-GHz purely phase-modulated analog signal, where the amplitude remains constant, with the modulation index varying from 1 to 64. As expected, the spectral bandwidth expands with increasing modulation index. This property allows phase modulation to gain SNR from spectral expansion, approximately 6.02 dB for each doubling of bandwidth[54]. When a 64-GBd PAM signal (32.32-GHz bandwidth) is further combined with the phase-modulated signal in the time domain to realize amplitude-phase layered modulation, their spectra become convolved in the frequency domain, as illustrated in Fig. 4b. As a result, the overall signal bandwidth is broadened, while most of the signal power remains concentrated in the lower-frequency region. The corresponding optical spectra after electro-optical modulation with different modulation indices are shown in Fig. 4c. The high-frequency roll-off of the optical and electrical components is compensated using the built-in de-emphasis functionality of the integrated coherent driver modulator. Finally, Fig. 4d shows the spectrum of the composite signal consisting of 64-GBd PAM and 8-GHz 1024-QAM modulation, along with the pilot tone. For comparison, the spectrum without the pilot tone is also provided. The modulation index is set to 12, and the pilot tone power is 10 dB lower than that of the converged signal.

First, we focus on the reception of the unipolar PAM-modulated digital signal at the ONU side. Figure 4e compares the BER performance of digital signals with and without the presence of phase-modulated analog signals, while the corresponding distributions of the recovered digital symbols are illustrated in the right. The negligible performance difference confirms that phase modulation has little influence on digital transmission. This is primarily because digital demodulation relies on the absolute-value operation

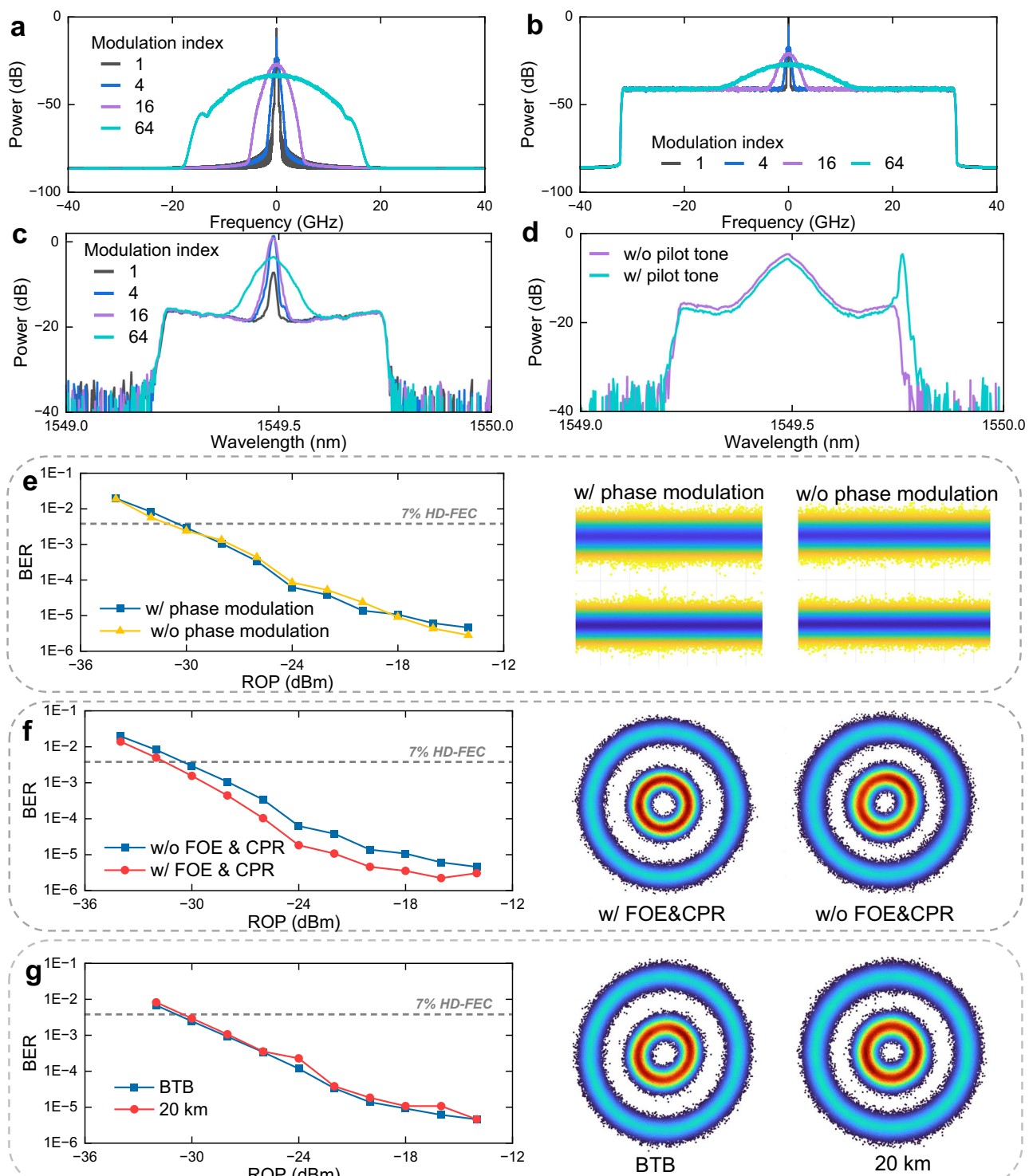

**Fig. 4 | Digital transmission performance over 20-km fibers. a** Digital spectra of phase-modulated analog signals without amplitude modulation. **b** Digital spectra with amplitude modulation. **c** Optical spectra of composite signals with different modulation indices. **d** Optical spectra with and without the pilot tone. **e** BER versus ROP for digital transmission with and without phase modulation. **f** BER versus ROP for digital transmission with and without FOE and CPR at the ONU side. **g** BER performance for back-to-back and 20-km scenarios.

to recover the signal amplitude, thereby discarding the phase information. Next, we compare the BER performance of DSP with and without FOE and CPR, as shown in Fig. 4f, with the corresponding recovered constellations depicted in the right inset. The results indicate that BER with FOE and CPR is slightly better than that without these procedures. This phenomenon can be explained by two factors: (1) phase ambiguity can degrade equalization

performance, introducing residual inter-symbol interference[13]; and (2) chromatic-dispersion-induced pulse broadening can be partially mitigated during equalization, which, however, also converts phase noise into amplitude noise[57]. Figure 4g presents a comparison of the ROP sensitivity for the proposed modulation scheme in back-to-back and 20-km transmission scenarios. The results indicate that fiber impairments, including CD

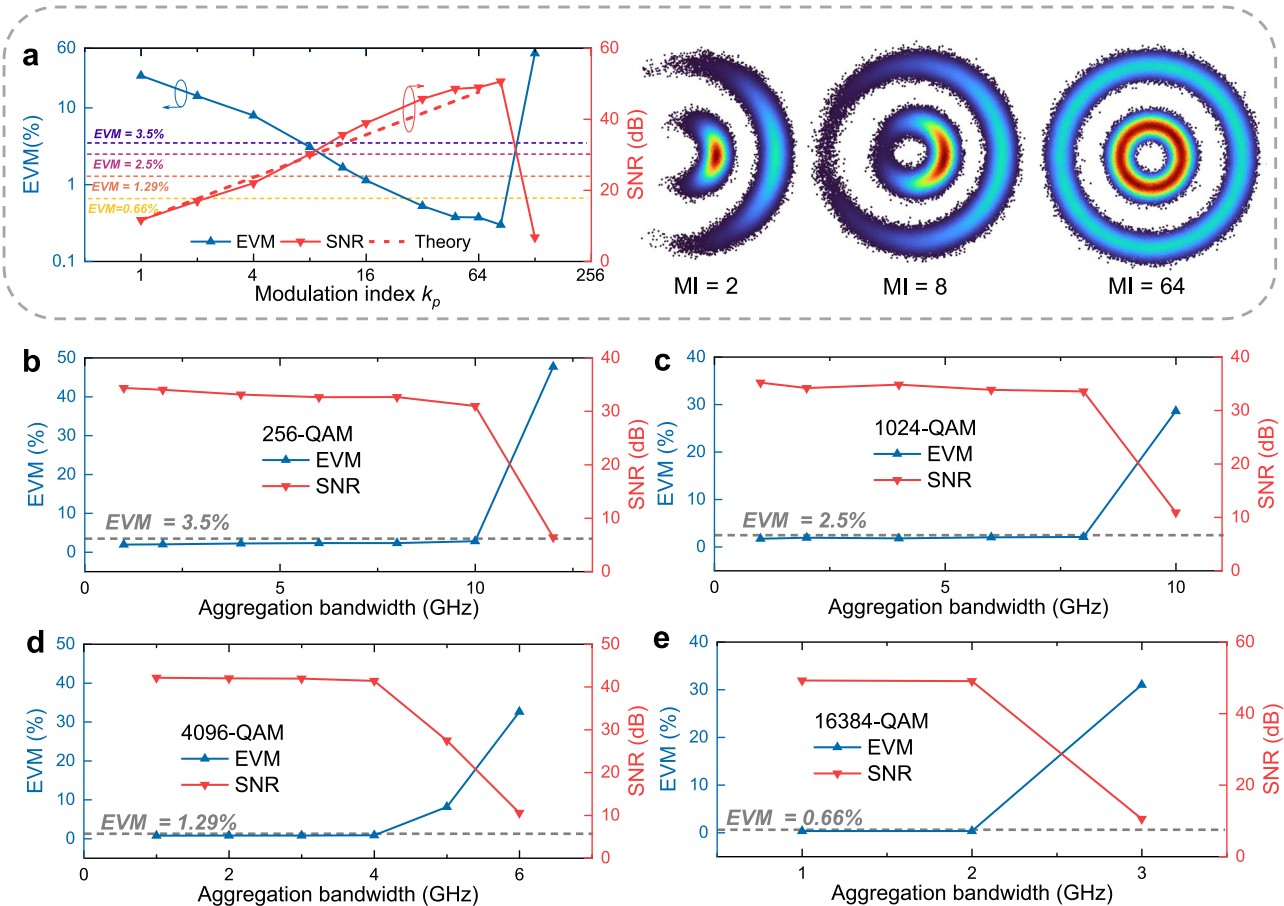

**Fig. 5 | Modulation index optimization and achievable aggregation bandwidth over 20-km fibers. a** EVM (left *Y*-axis) and SNR (right *Y*-axis) versus modulation index for a 1-GHz 1024-QAM phase-modulated signal, with recovered constellations. EVM (left *Y*-axis) and SNR (right *Y*-axis) as a function of aggregation bandwidth for 256-QAM (**b**), 1024-QAM (**c**), 4096-QAM (**d**), and 16384-QAM (**e**).

and polarization mode dispersion, introduce slight system degradation, resulting in a minor BER penalty.

Subsequently, we turn our attention to analog signal reception at the RRU side. The experiment is conducted to maximize the achievable aggregation bandwidth (i.e., the bandwidth of the OFDM signal prior to phase modulation) for each QAM order under the constraint of satisfying the required EVM and SNR. To this end, the modulation index is swept to assess the achievable EVM/SNR as a function of modulation index, as illustrated in Fig. 5a for a 1-GHz 1024-QAM phase-modulated signal. For analog demodulation, FOE and CPR are required to compensate for phase variations. With the modulation index increasing from 1 to 64, the SNR gain due to spectral expansion rises from 11.7 to 48.9 dB. When the modulation index is doubled, the measured SNR gain is about 6.2 dB, which is in good agreement with the theoretical 6.02 dB predicted by the quadratic scaling of phase modulation. The small excess is likely due to practical imperfections, such as a slightly narrower effective noise bandwidth introduced by filtering, colored noise from the transceiver front-ends, subtle imperfections in the transceiver response, and statistical uncertainty in SNR estimation. The corresponding recovered ring-shaped constellations for modulation index values of 2, 8, and 64 are presented in the right inset. Based on the 3GPP technical specifications and the derivation in refs. 31,58, we then determine the appropriate modulation indices for different QAM formats. Specifically, the EVM (SNR) requirements are 3.5% (29.1 dB) for 256-QAM, 2.5% (32 dB) for 1024-QAM, 1.29% (37.8 dB) for 4096-QAM, and 0.66% (43.6 dB) for 16384-QAM. Accordingly, the modulation indices are experimentally optimized to 11, 12, 22, and 44 for 256-QAM, 1024-QAM, 4096-QAM, and 16384-QAM, respectively. These values ensure that the

EVM remains below the corresponding 3GPP-specified thresholds while providing sufficient margin to accommodate increased aggregation bandwidth. The additional margin is necessary because wider signal bandwidths degrade the SNR at higher frequencies due to the transceiver's limited bandwidth and elevated noise floor. The EVM performance as a function of aggregation bandwidth for different QAM formats is presented in Fig. 5b–e. When the aggregation bandwidth exceeds its allowable limit, the composite signal spectrum extends beyond the system's Nyquist bandwidth, resulting in severe system impairment and a sharp degradation in SNR. Based on the EVM thresholds, the maximum achievable aggregation bandwidths for each polarization are determined to be 10 GHz, 8 GHz, 4 GHz, and 2 GHz for 256-QAM, 1024-QAM, 4096-QAM, and 16384-QAM, respectively.

Finally, we evaluate the optical power budget for both digital and analog transmissions. The results for different QAM formats are shown in Fig. 6a–c. Figure 6a shows the BER performance of the amplitude-modulated digital signal versus ROP when co-transmitting phase-modulated analog fronthaul signals with different QAM formats. Considering a 7% hard-decision forward error correction (HD-FEC) with a BER threshold of $3.8 \times 10^{-3}$, the required ROP for ONUs should exceed −30 dBm. With a launch optical power of 10.6 dBm, the maximum achievable optical power budget for PON is approximately 40.6 dB. Minor variations in optical power budget across different analog waveforms are attributable to experimental noise fluctuations and measurement repeatability, confirming the negligible crosstalk from the phase-modulated signal. Figure 6b, c respectively represent the EVM and SNR of the recovered phase-modulated analog waveforms versus ROP for different QAM formats. The saturation in EVM and SNR observed at higher ROPs is attributed to the inherent electrical

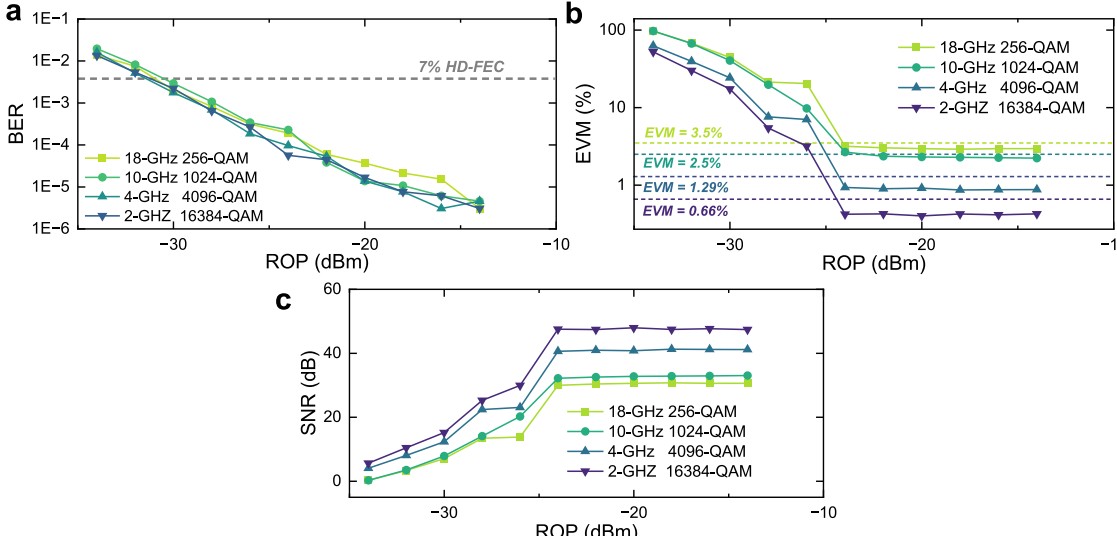

**Fig. 6 | ROP sensitivity for ONUs and RRUs over 20 km of SMF. a** BER of the amplitude-modulated digital bits versus ROP when carrying phase-modulated analog fronthaul signals with different QAM formats. **b** EVM of the recovered

phase-modulated analog waveforms versus ROP for different QAM formats. **c** SNR of the recovered phase-modulated analog waveforms versus ROP for different QAM formats.

SNR floor of the transceiver components. For analog waveforms, an ROP of −22 dBm ensures that all EVMs across different modulation formats meet their respective requirements, corresponding to an optical power budget of 32.6 dB for the RRUs.

### Field trial of integrated fixed-mobile access networks

In this section, we further assess the feasibility of the proposed integration scheme via transmission experiments over a field-deployed fiber link in Hong Kong, spanning up to 109 km. These results confirm its suitability for long-reach PON applications[59] and, more importantly, demonstrate its robustness against real-world impairments over extended distances, such as polarization fluctuations, and mechanical vibrations—that are difficult to fully replicate in a controlled laboratory environment. Note that, in practical deployments, the fiber length allocated to latency-sensitive mobile fronthaul can be limited to within 20 km while longer reaches can be reserved for PON services.

The transceiver configurations are identical to those used in the previous setup, and thus, we focus primarily on the field-deployed fiber link here. This link consists of four segments of SMF, totaling approximately 109 km: a 20 km segment (7 dB loss) from Hong Kong Polytechnic University (PolyU) to a data center in Tseung Kwan O; a 28.7 km segment (14 dB loss) from Tseung Kwan O to City University of Hong Kong (CityU); a 33.2 km segment (12 dB loss) from CityU to a data center in Chai Wan; and a 26.9 km segment (8.4 dB loss) from Chai Wan back to PolyU. The routes of these fiber segments are illustrated in Fig. 7a. The total link loss of 41.4 dB substantially exceeds the expected value for standard SMF with a loss of 0.2 dB/km, primarily due to numerous splice losses accumulated along the field-deployed route. In this field demonstration, conducted as a prototype validation of the proposed scheme, a 20-dB gain EDFA is inserted at an intermediate node, and a VOA is placed at the receiver side. These components serve as experimental tools to facilitate parameter sweeping and accurate measurement of the optical power budget. The EDFA, employed solely for experimental measurement purposes, compensates only for the excess splice and insertion losses in the deployed fiber link and does not alter the intrinsic optical power budget characteristics of the proposed modulation scheme itself. In conventional PON deployments with cleaner fiber infrastructure and fewer splices, such an EDFA would not be required. This setup enables thorough validation of the scheme's robustness against field-induced polarization variations and phase fluctuations. The OLT transmitter and receiver for ONU and RRU are both housed within the PolyU laboratory. Relative to the DSP configuration adopted in the 20-km

case, the only additional step is CD compensation, which mitigates the accumulated dispersion over 109 km.

Similar to the preceding experimental procedures, we characterize the SNR or EVM as a function of modulation index, as shown in Fig. 8a, to determine the optimal value for different QAM formats. Consistent with the 20 km case, the modulation indices are set to 11, 12, 22, and 44 for 256-QAM, 1024-QAM, 4096-QAM, and 16384-QAM, respectively. For 64-QAM, a modulation index of 6 is sufficient to meet its SNR requirement of 21.9 dB (corresponding to 8% EVM). Under these settings, we evaluate the maximum achievable aggregation bandwidth with different modulation formats, as displayed in Fig. 8b–f. The launch power is maintained at 10.6 dBm. These results show that analog signals with per-polarization bandwidths of 18 GHz, 10 GHz, 8 GHz, 4 GHz, and 2 GHz can be transmitted for 64-QAM, 256-QAM, 1024-QAM, 4096-QAM, and 16384-QAM, respectively. The corresponding ROP sensitivities for these aggregation bandwidths and modulation formats are shown in Fig. 9a–c. For the 109-km transmission scenario, an ROP of −28 dBm ensures that digital transmission remains below the 7% HD-FEC threshold, whereas analog transmission requires an ROP above −20 dBm to satisfy the EVM requirements for all modulation formats. The corresponding optical power budgets are 38.6 and 30.6 dB for the ONUs and RRUs, respectively. Under these conditions, the achieved total aggregated bandwidths with two polarizations, along with their corresponding CPRI-equivalent rates, are presented in Fig. 9d. The CPRI-equivalent rates are obtained by multiplying the aggregated bandwidth by 61.4 b/s/Hz[7,19]. A beyond-terabit-per-second CPRI-equivalent rate can be achieved using 64-QAM and 256-QAM, while for 1024-QAM, the rate reaches approximately 982 Gb/s. The measured EVMs and SNRs at an ROP of −20 dBm are shown as bars in Fig. 10a, and the recovered constellations for 64-QAM, 256-QAM, 1024-QAM, 4096-QAM, and 16384-QAM are illustrated in Fig. 10b–f. The clear constellation diagrams confirm the effectiveness of SNR adaptation in the amplitude-phase layered modulation scheme. Specifically, we achieve an SNR of 23.1 dB with an EVM of 7% for 64-QAM, 29.8 dB with 3.25% for 256-QAM, 32.3 dB with 2.42% for 1024-QAM, 40.5 dB with 0.95% for 4096-QAM, and 47.2 dB with 0.43% for 16384-QAM. These results further demonstrate the feasibility of applying the proposed amplitude-phase layered modulation method to metro-access optical network convergence.

### Discussion

Table 1 summarizes the state-of-the-art experiments on integrated digital and analog signal transmission systems. The comparison

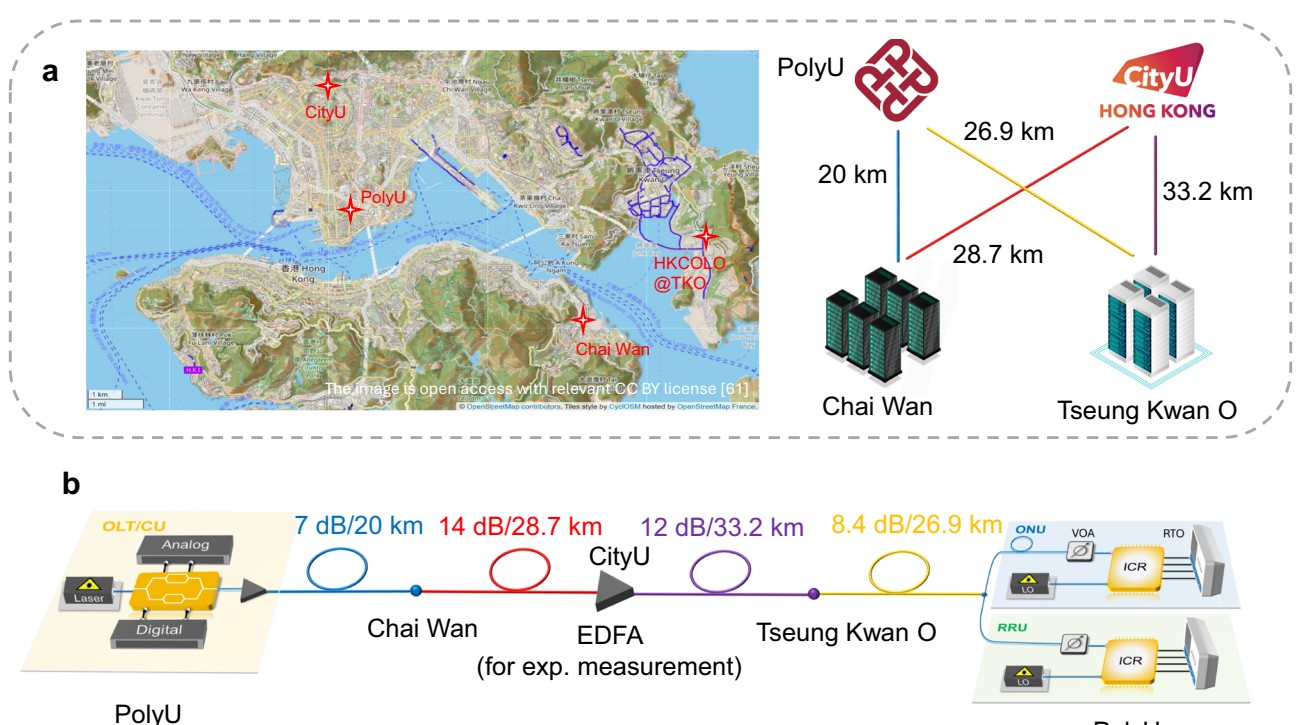

**Fig. 7 | Field trial of integrated fixed-mobile access networks. a** Field-deployed fiber link layout in Hong Kong[61]. The 109-km single-mode fiber comprises four strands connecting PolyU, CityU, and two data centers in Chai Wan and Tseung Kwan O. **b** Experimental setup for the field trial.

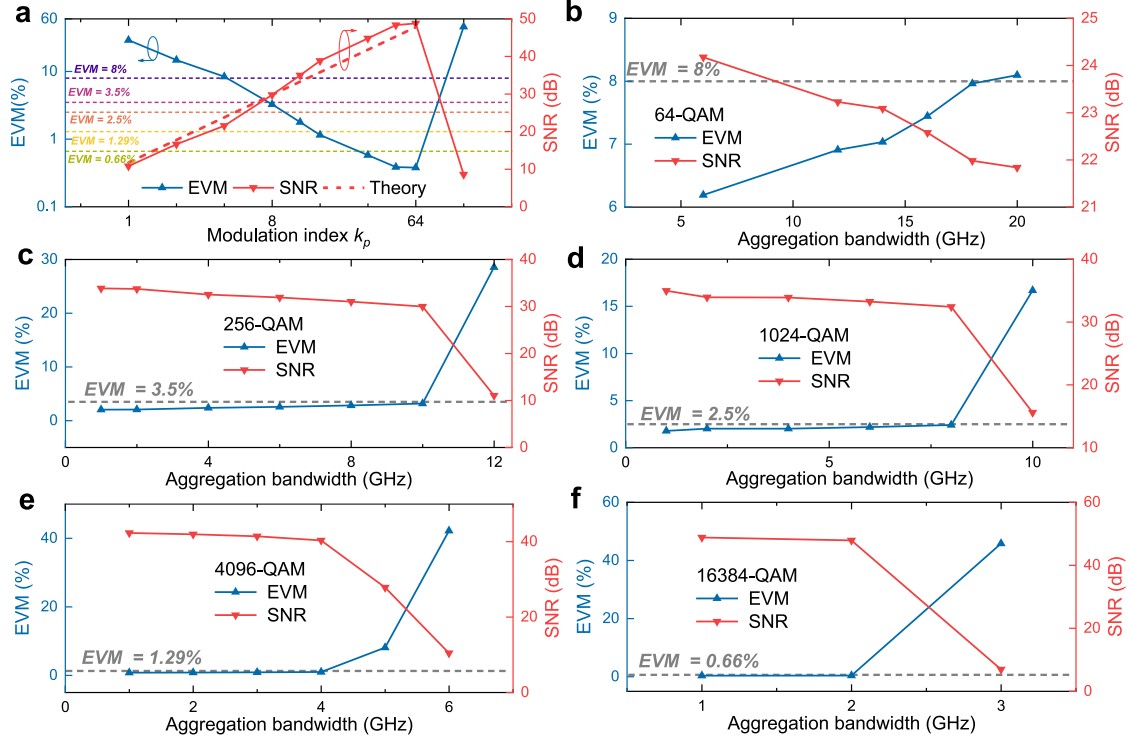

**Fig. 8 | Modulation index optimization and achievable aggregation bandwidth over 109-km field-deployed fibers. a** EVM (left Y-axis) and SNR (right Y-axis) versus modulation index for a 1-GHz 1024-QAM phase-modulated signal. EVM (left Y-axis) and SNR (right Y-axis) as a function of aggregation bandwidth for 64-QAM (**b**), 256-QAM (**c**), 1024-QAM (**d**), 4096-QAM (**e**), and 16384-QAM (**f**).

includes key metrics such as achieved analog aggregation bandwidth, CPRI-equivalent rate, QAM order, SNR, multiplexing technique, digital data rate, transmission distance, receiver detection method, and demonstration environment. Notably, our system achieves

network convergence with the highest CPRI-equivalent rate and the largest modulation order demonstrated over a 109-km field-deployed fiber link. Moreover, the proposed modulation method for surpassing the SNR ceiling is inherently scalable: by increasing the modulation

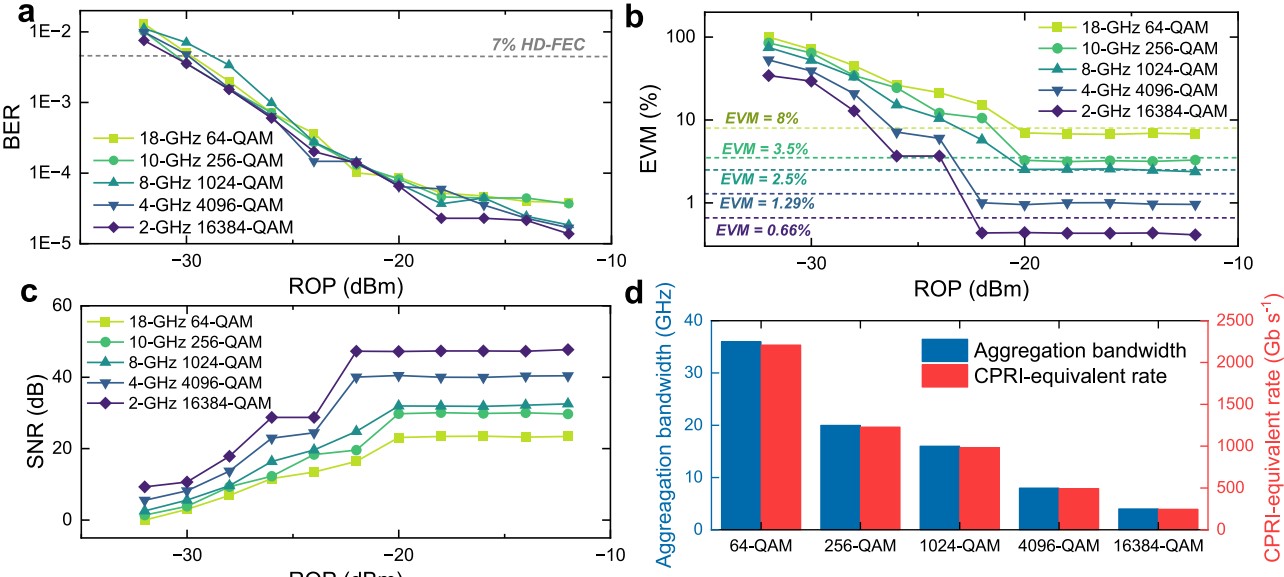

**Fig. 9 | ROP sensitivity for ONUs and RRUs, and maximum achievable CPRI-equivalent rate over 109 km of SMF. a** BER of the amplitude-modulated digital bits versus ROP when carrying phase-modulated analog fronthaul signals with different QAM formats. **b** EVM of the recovered phase-modulated analog waveforms versus ROP for different QAM formats. **c** SNR of the recovered phase-modulated analog waveforms versus ROP for different QAM formats. **d** Achievable total dual-polarization aggregation bandwidth (blue, left Y-axis) and CPRI-equivalent rate (red, right Y-axis).

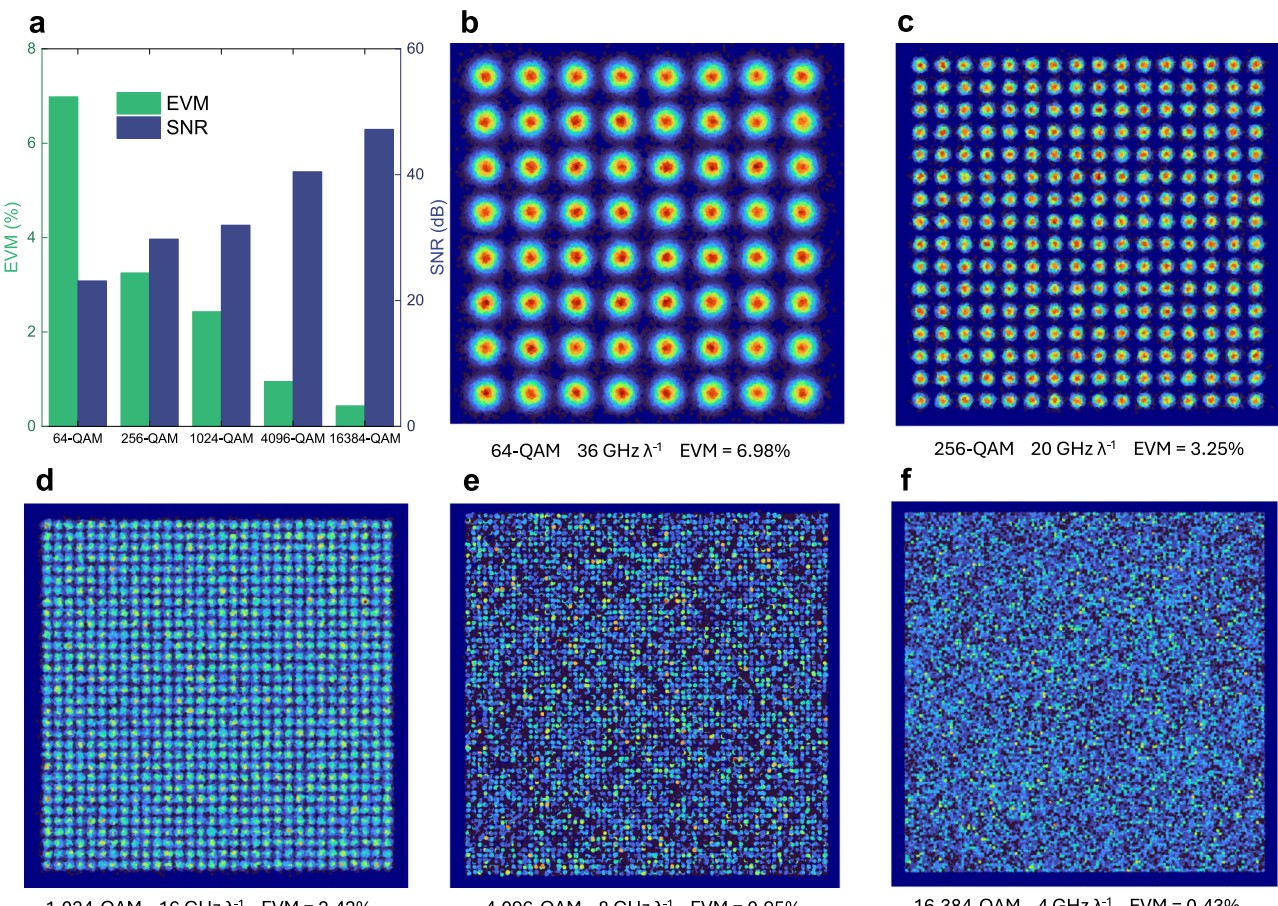

**Fig. 10 | Achieved SNR and EVM, and recovered constellations for different modulation formats. a** EVM (left Y-axis) and SNR (right Y-axis) versus modulation format. Recovered constellations for 64-QAM with 36 GHz aggregation bandwidth (**b**), 256-QAM with 20 GHz (**c**), 1024-QAM with 16 GHz (**d**), 4096-QAM with 8 GHz (**e**), and 16384-QAM with 4 GHz (**f**).

**Table 1 | Comparison of integrated digital and analog transmission for converged access networks**

| Reference | 40 | 38 | 49 | 45 | 50 | 48 | 44 | 46 | This work |
|---|---|---|---|---|---|---|---|---|---|
| Aggregation bandwidth (GHz) | 1.5 | 0.48 | 0.5 | 18.36 | 3.4 | 4 | 2.4 | 6 | 36/20/16/8/4 |
| CPRI-equivalent rate (Gb/s) | 92 | 29 | 31 | 1127 | 208 | 245 | 147 | 369 | 2210/1228/982/491/246 |
| QAM Order | 16 | 64 | 64 | 16 | 64 | 64 | 128 | 64 | 64/256/1024/4096/16384 |
| SNR of Analog Signals (dB) | 18.2 | 18.5 | N/A | N/A | 25.8 | 23 | 26 | 24 | 23.1/29.8/32.3/40.5/47.2 |
| Multiplexing technique | WDM | WDM | WDM | WDM&PDM | FDM | FDM | FDM | Chirp-based FDM | Amplitude-phase-layered modulation |
| Digital datarate (Gb/s) | 400 | 252 | 40 | 70 | 45 | 56 | 2.5 | 64 | 128 |
| Transmission distance (km) | 50 | 49 | 20 | 20 | 10 | 25 | 25 | 10 | 109 |
| Detection method | Coherent & DD | Coherent & DD | Coherent | DD | DD | DD | DD | DD | Coherent |
| Environment | Lab | Lab | Lab | Lab | Lab | Lab | Lab | Lab | Field |

*WDM* Wavelength Division Multiplexing, *PDM* Polarization Division Multiplexing, *FDM* Frequency Division Multiplexing, *DD* Direct Detection.

index, even higher-order modulation formats can be supported with excellent signal fidelity.

This amplitude-phase layered modulation offers several advantages for the integration of digital and analog transmission. For the digital channel, information is carried on the unipolar optical amplitude, making it inherently phase-insensitive and thus tolerant to channel impairments such as frequency offset and laser phase noise. This property enables a simplified DSP architecture at the receiver, eliminating the need for conventional FOE and CPR blocks in coherent detection. As a result, the associated power consumption and processing latency are significantly reduced, with only a negligible penalty in transmission performance. However, it should also be noted that the achievable data rate for the digital transmission is approximately halved compared to full IQ modulation on both polarizations. This represents a trade-off: simplified DSP complexity and enhanced flexibility for hybrid digital-analog transmission are achieved at the expense of reduced bandwidth efficiency. For the analog channel, the PM enables the SNR scaling by increasing the modulation index. The SNR will be improved by 6.02 dB per doubling the modulation index, thus improving the fidelity of analog signals. By adjusting the modulation index, the analog transmission could break through the limit of the inherent channel's SNR and support high-order modulation formats.

The proposed amplitude-phase layered modulation scheme is primarily designed for downstream transmission in integrated fixed-mobile access networks. It remains fully compatible with existing upstream architectures, where conventional access networks typically employ distinct wavelengths to mitigate Rayleigh backscattering in bidirectional fiber transmission[10]. Looking forward to next-generation fully converged fixed-mobile upstream transmission, digital signals from ONUs for fixed access and analog signals from RRUs for mobile fronthaul can be multiplexed either in different time slots on the same wavelength, or in the same time slot using adjacent wavelengths. These time- and wavelength-domain multiplexing schemes[60] enable simultaneous reception of both signal types with a single coherent receiver at the OLT, offering a promising path toward reduced hardware complexity. Investigating such upstream integration strategies constitutes an important and exciting direction for future research.

To address cost and power consumption concerns, the proposed scheme can also be implemented in IM-DD systems, making it compatible with current access networks that predominantly employ cost-effective optics such as directly-modulated lasers and electro-absorption modulated lasers. In this alternative approach, digital subcarrier multiplexing is used: the converged complex-valued baseband signal is digitally frequency-shifted to an intermediate frequency (IF)[28], and only the real part of the shifted signal is taken to drive the intensity modulator. In this way, only intensity modulation is required at the transmitter. Furthermore, the receiver can be further simplified to a single photodetector followed by envelope detection. A digital hybrid is then employed to down-convert the IF signal to baseband, thereby further reducing hardware complexity and cost in this IM-DD implementation.

However, from a hardware compatibility perspective, the proposed scheme is better aligned with existing intradyne coherent systems that employ full IQ modulation, facilitating seamless integration with commercially available optical modules. Nevertheless, coherent optics is increasingly being considered for next-generation short-reach optical networks—including data center interconnects[13], PON[12], and mobile fronthaul[19]—despite the associated higher DSP power consumption and hardware complexity. This trend is driven by the advantages of coherent detection in supporting multi-dimensional modulation and providing superior robustness against optical impairments. Furthermore, to improve hardware efficiency in the proposed scheme, the transmitter can be further simplified by employing separate intensity and phase modulators, thereby reducing complexity while maintaining the layered modulation functionality. In this setup, the digital PAM signal drives the intensity modulator to generate optical fields with varying amplitudes, while the analog signal directly drives the phase modulator. The modulation index is adjusted via a gain-controlled electrical driver placed before the phase modulator, thereby reducing signal-processing latency at the transmitter. This simplified configuration enables the coexistence of digital and analog modulation across different CUs[40,51], supporting flexible add-drop functionality and offering clear advantages for latency-sensitive access network applications.

## Conclusions

We demonstrate a converged fixed–mobile coherent optical access network operating over a single optical carrier by developing an amplitude–phase layered modulation scheme. We successfully achieve the co-delivery of a 128 Gb/s digital signal and wireless analog signals with various QAM formats, and dual-polarization aggregated bandwidths ranging from 36 GHz (64-QAM) to 4 GHz (16384-QAM), over a 109-km operator-deployed telecom fiber network in Hong Kong. The proposed scheme eliminates conventional DSP operation—such as FOE and CPR—in a digital coherent system, thereby reducing power consumption and processing latency for ONUs, while achieving a power budget exceeding 38.6 dB. For analog transmission, PM enables SNR scaling from 23.1 to 47.2 dB, supporting a single-wavelength CPRI-equivalent rate of 1.2 Tb/s with 256-QAM. These results highlight the potential of amplitude-phase layered modulation to enable efficient, low-complexity convergence of fixed and mobile access services.

Looking ahead, the proposed integration scheme opens up several promising application scenarios. For instance, it can enable the co-delivery of PON services and indoor millimeter-wave signals to support fiber-to-the-room deployments, providing high-speed broadband access while simultaneously enhancing indoor wireless coverage through a unified infrastructure that reduces deployment cost and complexity. In addition, the scheme supports the joint transmission of digital CPRI signals and analog millimeter-wave signals for 4G and 5G fronthaul, enabling efficient utilization of fiber resources by allowing legacy and next-generation wireless services to share the same optical link. This coexistence enhances network flexibility and scalability while minimizing the need for additional fiber deployment. Overall, the proposed approach provides a promising pathway for supporting not only fixed-mobile optical access but also other integrated digital-analog communication paradigms, including visible light communications, free-space optics, and terahertz systems, thereby paving the way toward unified optical-wireless networks for beyond-5G and 6G communications.

## Data availability

The data supporting this study have been publicly deposited in the Zenodo database, https://doi.org/10.5281/zenodo.18079921. All other raw data generated or analyzed during this work are available from the corresponding authors upon request.

## Code availability

The code supporting this study has been publicly deposited in the Zenodo database,https://doi.org/10.5281/zenodo.18079921.

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

## Acknowledgements

This work was supported by the National Key Research and Development Program of China (2024YFB2908001, 2024YFB2807703), the Hong Kong Government General Research Fund (PolyU25227524, PolyU15225423, C5078-24G), and Hong Kong Polytechnic University (1-BEA9, G-SB1P, 1-CD8L,1-CDM9). We are grateful for the generous donation of the deployed fiber links by HKCOLO.

## Author contributions

Q.Wu conceived the idea. Q.Wu, X.Y.Zhang, and H.Q.Wei conducted the experiments and performed the initial data analysis. Q.Wu and Z.X.Wei led the writing of the manuscript. X.Y.Zhang, H.Q.Wei, J.H.Huo, X.R.Huang, T.H.Ji, Z.P.Xu, C.Lu, A.P.T.Lau, and K.P.Zhong contributed to the modeling, experiment design, interpretation of the results, and refinement of the manuscript through discussions and review.

## Competing interests

The authors declare no competing interests.
