## [Transparent Peer Review file · Communications Engineering]

Integrating fixed and mobile coherent optical access networks for unified broadband services

Corresponding Author: Dr Qi Wu

Version 0:

Reviewer comments:

Reviewer #1

(Remarks to the Author)

In this paper, the authors present a novel concept for integrating fixed and radio access networks using a shared infrastructure, including transceivers. They demonstrate the feasibility of a layered modulation scheme based on both phase and amplitude, where each dimension is used to provide a distinct connectivity service. The idea is original and timely, especially considering the growing adoption of coherent transceivers and their increasing attention from standardization bodies in the access segment.

The results are highly relevant, representing a significant advancement in the state of the art. They include not only laboratory experiments but also data obtained from field trials. Moreover, the authors have made both data and code publicly available in the Zenodo repository, which greatly supports reproducibility.

That said, the manuscript is challenging to follow due to its weak structural organization. I recommend adopting a clearer and more homogeneous structure. Specifically, Section 2 should focus on the concept and methods, providing a proper description of the approach before presenting the experiments, results, and discussion. Accordingly, I suggest reorganizing the paper as follows: Section 2 as the "Methods" section (including current Sections 2.1 and 4), Section 3 as the "Results" section, and Section 4 devoted to the discussion.

Finally, a minor comment concerns the notation used for capacity units. In the research community, capacity is typically expressed as b/s, Gb/s, or Tb/s, whereas the manuscript consistently uses $b\cdot s^{-1}$, $Gb\cdot s^{-1}$, or $Tb\cdot s^{-1}$. Unless there is a specific reason for this choice, I recommend adopting the standard notation b/s, Gb/s, or Tb/s.

Reviewer #2

(Remarks to the Author)

This paper is an extended version of an ECOC 2025 post-deadline paper. The key of the work is the use of intensity modulation for usual PON and phase modulation for Fronthaul transport, using coherent receivers..

The introduction reminds the current status of ITU-T PONs, and mobile access (such as C-RAN, fronthaul, A-RoF). The introduction is generally well detailed, with many references from various sources.

The results are provided in several phases in section 2. First, the concepts are briefly introduced, before moving to the lab experiments, and then to the field trial in Hong Kong area. This section is generally clear, while some comments are listed below. It is surprising to see the "method" only in Section 4 however. It should appear before the setup and results.

Section 3 repeats many elements of previous sections and should be rewritten to avoid such repetitions. For example, the first sentences are an introduction of what was actually presented in sections 1 and 2. The discussion compare the author's work to the state of the art, but does not talk about access networks and its very low-cost specificities. Currently, no MZM (not IQ MZM) are used in access networks, because they are too expensive. And I am not mentioning DSP. Also, the paper does not discuss the upstream communication, which is more challenging, since the tree topology generally imposes even lower costs at ONU / RU side. There is no discussion about the relevance of Fronthaul over 109 km.

Generally speaking, the paper is clear, while some sections need to be reconfigured to appear in the right place.

The following lists more specific points:

- Page 2: when introducing eCPRI, please specify that it is standard and relies on Ethernet protocol.
- Page 2: "Such inefficiency makes it impractical to accommodate the terabit-per-second rates envisioned for 6G networks [19]. ». [19] is a paper about Fronthaul using electro-optic combs. Maybe a more industry-related paper would be more appropriate (standards from 3GPP, ORAN alliance, ... or even a paper, but on 6G).
- Page 3: "operators are moving toward unified architectures that combine fixed and mobile access [32–34]... [...] allows for shared use of existing optical fiber infrastructure. ». Operators tend to share their active equipment, such as OLTs. But the fiber infrastructure is generally not shared. You do not want an FTTH operator to unplug an antenna site by mistake. Please rephrase.
- P6 fig 2: Is the "OLT" emitter using dual polarization? or single polarization as represented?
- Page 7: the experimental setup should be described in more details: gain of the EDFA, wavelength of the LOs, linewidth of the LOs, insertion losses of the IQ MZM.
- Page 7: what is the frequency offset between emitters and LOs?
- P 11 fig 5: a few points are out of scale.
- P 11 fig 5: figure is difficult to read as the 2nd y-axis mixes EVM (%) and SNR (dB). Maybe the solution would be to propose a figure with BER, another with the four EVMs, and another with the 4 SNRs.
- P 10: could the authors comment on the 109 km reach? usually, eCPRI is latency sensitive and thus limited to 20 km reach typically. Otherwise, latency is too high from Fronthaul and low-layer splits.
- P 13 fig 8 : similar comments than those to fig 5
- Page 15: author affirm that modulations such as 65536 QAM can be supported, without supporting that claim. The experiments go as high as 16384 QAM.
- Page 17: a reference is needed beside Carson's rule
- There is no proper "conclusion" section in the paper.

Typos:

- Bibliography: capital letters vanished in some references, such as in [11]. The "g" of "100g" should be capital. Please check all references.

Reviewer #3

(Remarks to the Author)
Please see attachment.

Reviewer #4

(Remarks to the Author)

I co-reviewed this manuscript with one of the reviewers who provided the listed reports. This is part of the Communications Engineering initiative to facilitate training in peer review and to provide appropriate recognition for Early Career Researchers who co-review manuscripts.

Reviewer #5

(Remarks to the Author)

This is a well written review paper that is an extension of the authors' previous ECOC post-deadline paper in which they proposed a digital/analog modulation scheme to support fixed/mobile optical fronthaul access. My comments mainly concern clarification of their results.

1. The phase-modulated system uses OFDM, which has parameters such as number of carriers, guard interval, etc. The authors should include details of the OFDM implementation.

2. In Figs. 5 and 8, the authors opted to evaluate performance using BER for the amplitude-modulated digital system, but EVM for the phase-modulated "analog" system which actually represents digital mobile data. Why the difference? BER should be a superior metric for system performance, as BER can only be accurately inferred by EVM when noise is Gaussian, which is not guaranteed or proven in the paper. Since the authors generated phase-modulated OFDM signals containing digital data, it should not be difficult to just estimate BER? At the very least, the users should justify the use of EVM rather than BER.

3. In Figs. 4 and 7, the authors first performed a sweep of EVM/SNR versus modulation index to find the k_p necessary to support various QAM formats, and then swept EVM/SNR against the aggregation bandwidth of the OFDM signal to find the maximum bandwidth that can be supported. The thresholds for each modulation format is a bit hard to see. Maybe the authors can add dotted lines for the threshold EVM (if this is even an acceptable metric, or else BER can be used) so we can see where the blue lines cross the threshold.

Also, the two vertical axes may be redundant. I believe they show the same information as $EVM = 1/SNR$. If this is correct, the authors should state this relationship in the paper.

4. Likewise, it would help the reader to show dashed lines for the BER threshold and minimum ROP in Figs. 3(e-g), 5(a-d) and 8(a-d) for results of amplitude-modulated system.

5. Figs. 5 and 8, which combine results for both the amplitude-modulated and phase-modulated systems, seems cluttered. The right vertical axes are also confusing as they show EVM (%) and SNR (dB) on the same scale, but they are completely different things. If EVM and SNR are inverses of each other, maybe you don't need to show both. Furthermore, if the phase-modulated system was evaluated using BER rather than EVM/SNR, the two sets of BERs can be shown on one vertical axis and would be easier to read.

I observe saturation in the EVM/SNR of the phase-modulated system. I assume this is due to transceiver SNR. This should be discussed and stated in the paper.

6. On page 14, "...bandwidths ranging from 36 GHz (64-QAM) to 4 GHz (16384-QAM)". It should be noted that these number are a summation of the bandwidths of both polarizations.

7. In the introduction and in Fig. 8(e), the authors cited a number of 61.4 bit/s to represent 1 Hz of wireless data. Where does this number come from? I cannot find this number in the cited references. CPRI samples and digitizes the analog waveforms of the wireless signal (with additional overhead). The ratio between the required CPRI data rate vs wireless data rate should not be a constant. Furthermore, if the aggregation bandwidth was X, then in case CPRI was used, the achievable wireless data rate Y would be much lower than X due to a low spectral efficiency. The correct comparison should be the data rate achieved ($X \text{ Gbaud} * B \text{ bits/symbol}$) with the proposed method versus data rate Y which would be achieved using CPRI.

Version 1:

Reviewer comments:

Reviewer #1

(Remarks to the Author)

The authors have successfully addressed all comments raised. Thank you!

Reviewer #2

(Remarks to the Author)

The authors addressed the requested revisions, doing a great effort about the paper's structure. I have been looking point by point at the response letter, and the answers are correct and satisfying.

I feel that the issues points raised in the previous round of review have been satisfactorily addressed.

Reviewer #3

(Remarks to the Author)

The authors have provided a revised manuscript. All concerns have been addressed satisfactorily. I thank the authors for the modifications and their detailed response letter.

Reviewer #4

(Remarks to the Author)

I co-reviewed this manuscript with one of the reviewers who provided the listed reports. This is part of the Communications Engineering initiative to facilitate training in peer review and to provide appropriate recognition for Early Career Researchers who co-review manuscripts.

Reviewer #5

(Remarks to the Author)

My previous comments have been answered.

Response to Reviewers' Comments

Response to Reviewer 1

Comment:

In this paper, the authors present a novel concept for integrating fixed and radio access networks using a shared infrastructure, including transceivers. They demonstrate the feasibility of a layered modulation scheme based on both phase and amplitude, where each dimension is used to provide a distinct connectivity service. The idea is original and timely, especially considering the growing adoption of coherent transceivers and their increasing attention from standardization bodies in the access segment.

The results are highly relevant, representing a significant advancement in the state of the art. They include not only laboratory experiments but also data obtained from field trials. Moreover, the authors have made both data and code publicly available in the Zenodo repository, which greatly supports reproducibility.

Response:

We sincerely thank the reviewer for the valuable comments. We have carefully studied the comments and have made revisions accordingly. All changes made in the revised version are coloured in blue.

Comment 1:

That said, the manuscript is challenging to follow due to its weak structural organization. I recommend adopting a clearer and more homogeneous structure. Specifically, Section 2 should focus on the concept and methods, providing a proper description of the approach before presenting the experiments, results, and discussion. Accordingly, I suggest reorganizing the paper as follows: Section 2 as the “Methods” section (including current Sections 2.1 and 4), Section 3 as the “Results” section, and Section 4 devoted to the discussion.

Response:

We thank the reviewer for this valuable suggestion regarding the structural organization of the manuscript. To improve clarity and readability, we have reorganized the paper

in the revised version largely according to the reviewer’s recommendations. Specifically, the proposed concept and all methodological details are now consolidated in a dedicated Section 2 titled “Concepts and Methods”. The experimental results are presented in Section 3 (“Results”), and an in-depth discussion is provided in Section 4 (“Discussion”). Additionally, we have included a new Section 5 (“Conclusion”) to summarize the key findings and outline future research directions.

Comment 2:

Finally, a minor comment concerns the notation used for capacity units. In the research community, capacity is typically expressed as b/s, Gb/s, or Tb/s, whereas the manuscript consistently uses $\text{b}\cdot\text{s}^{-1}$, $\text{Gb}\cdot\text{s}^{-1}$, or $\text{Tb}\cdot\text{s}^{-1}$. Unless there is a specific reason for this choice, I recommend adopting the standard notation b/s, Gb/s, or Tb/s.

Response:

We thank the reviewer for this suggestion. In the revised manuscript, we have unified the capacity units throughout the paper to the standard notation (b/s, Gb/s, and Tb/s).

Response to Reviewer 2

Comment:

This paper is an extended version of an ECOC 2025 post-deadline paper. The key of the work is the use of intensity modulation for usual PON and phase modulation for Fronthaul transport, using coherent receivers. The introduction reminds the current status of ITU-T PONs, and mobile access (such as C-RAN, fronthaul, A-RoF).

The introduction is generally well detailed, with many references from various sources. The results are provided in several phases in section 2. First, the concepts are briefly introduced, before moving to the lab experiments, and then to the field trial in Hong Kong area. This section is generally clear, while some comments are listed below. It is surprising to see the “method” only in Section 4 however. It should appear before the setup and results. Section 3 repeats many elements of previous sections and should be rewritten to avoid such repetitions. For example, the first sentences are an introduction of what was actually presented in sections 1 and 2. The discussion compare the author’s work to the state of the art, but does not talk about access networks and its very low-cost specificities. Currently, no MZM (not IQ MZM) are used in access networks, because they are too expensive. And I am not mentioning DSP. Also, the paper does not discuss the upstream communication, which is more challenging, since the tree topology generally imposes even lower costs at ONU / RU side. There is no discussion about the relevance of Fronthaul over 109 km.

Generally speaking, the paper is clear, while some sections need to be reconfigured to appear in the right place.

Response:

We sincerely thank the reviewer for the detailed and constructive feedback. We have carefully studied the comments raised by the reviewer and have made corrections accordingly. All changes made in the revised version are coloured in blue.

First, to improve the structural organization, all methodological details have been consolidated into a new Section 2 titled “Concepts and Methods,” placed before the experimental setup and results (Section 3). The original Discussion section has been split into Section 4 (“Discussion”) and a new Section 5 (“Conclusion”) to provide a clearer structure and a dedicated summary of key achievements and future outlook. In addition, we have rephrased the repetitions and eliminated redundancy with earlier sections.

As the reviewer correctly pointed out, access networks are highly cost-sensitive, resulting in the predominant use of low-cost intensity-modulated direct-detection systems based on directly modulated lasers (DMLs) and electro-absorption modulated lasers (EMLs), which also offer lower cost and power consumption. To address these cost concerns, we have added the following discussion in the Section 4 (Discussion) of the revised manuscript: To address cost and power consumption concerns, the proposed scheme can

also be implemented in IM-DD systems, making it compatible with current access networks that predominantly employ cost-effective optics such as directly-modulated lasers and electro-absorption modulated lasers. In this alternative approach, digital subcarrier multiplexing is used: the converged complex-valued baseband signal is digitally frequency-shifted to an intermediate frequency (IF) [28], and only the real part of the shifted signal is taken to drive the intensity modulator. In this way, only intensity modulation is required at the transmitter. Furthermore, the receiver can be further simplified to a single photodetector for envelope detection. A digital hybrid is then employed to down-convert the IF signal to baseband, thereby further reducing hardware complexity and cost in this IM-DD implementation.

In addition, there is a growing trend toward adopting coherent optics in short-reach scenarios. Coherent technology is increasingly being explored for next-generation applications, including data center interconnects [13], PONs [12], and mobile fronthaul [19], despite the higher DSP power consumption and hardware complexity. This shift is driven by the significant advantages of coherent detection, such as support for multi-dimensional modulation formats and enhanced robustness against optical impairments. We have incorporated this balanced discussion into the last paragraph of Discussion in the revised manuscript as follows: Nevertheless, coherent optics is increasingly being considered for next-generation short-reach optical networks—including data center interconnects [13], PON [12], and mobile fronthaul [19]—despite the associated higher DSP power consumption and hardware complexity. This trend is driven by the advantages of coherent detection in supporting multi-dimensional modulation and providing superior robustness against optical impairments.

A discussion about the upstream communication have been added in Discussion: The proposed amplitude–phase layered modulation scheme is primarily designed for downstream transmission in integrated fixed-mobile access networks. It remains fully compatible with existing upstream architectures, where conventional access networks typically employ distinct wavelengths to mitigate Rayleigh backscattering in bidirectional fiber transmission [10]. Looking forward to next-generation fully converged fixed–mobile upstream transmission, digital signals from ONUs for fixed access and analog signals from RRUs for mobile fronthaul can be multiplexed either in different time slots on the same wavelength, or in the same time slot using adjacent wavelengths. These time- and wavelength-domain multiplexing schemes [60] enable simultaneous reception of both signal types with a single coherent receiver at the OLT, offering a promising path toward reduced hardware complexity. Investigating such upstream integration strategies constitutes an important and exciting direction for future research.

The comments for the relevance of Fronthaul over 109 km has been added (See Comment 9 below).

Comment 1:

Page 2: when introducing eCPRI, please specify that it is standard and relies on Ethernet protocol.

Response:

We thank the reviewer for this valuable suggestion. To address this point and improve clarity, we have revised the description of eCPRI and added relevant references in the third paragraph of Section 1 (Introduction) of the revised manuscript as follows: On the mobile access side, the concept of centralized radio access network (C-RAN) has gained significant traction, where fronthaul technologies such as the common public radio interface (CPRI) [17] and its evolved successor, the enhanced CPRI (eCPRI) [18]—an industry-standard specification that supports efficient packet-based transport over Ethernet networks—serve as key enablers.

[R1] eCPRI Specification V2.0. [https://www.cpri.info/downloads/eCPRI v 2.0 2019 05 10c.pdf](https://www.cpri.info/downloads/eCPRI_v2.0_20190510c.pdf) (Ref. [18] in this manuscript)

Comment 2:

Page 2: “Such inefficiency makes it impractical to accommodate the terabit-per-second rates envisioned for 6G networks [19]. [19] is a paper about Fronthaul using electro-optic combs. Maybe a more industry-related paper would be more appropriate (standards from 3GPP, ORAN alliance, . . . or even a paper, but on 6G).

Response:

We thank the reviewer for this constructive suggestion. Accordingly, we have replaced reference [19] (now [21]) with the following article, which provides a comprehensive discussion of the performance targets and technological challenges for future 6G networks.

[R2] S. Chen, Y.-C. Liang, S. Sun, S. Kang, W. Cheng, and M. Peng, “Vision, requirements, and technology trend of 6G: How to tackle the challenges of system coverage, capacity, user data-rate and movement speed,” *IEEE Wireless Communications*, vol. 27, no. 2, pp. 218–228, Apr. 2020. (Ref. [21] in this manuscript)

Comment 3:

Page 3: “operators are moving toward unified architectures that combine fixed and mobile access [32–34]... [...] allows for shared use of existing optical fiber infrastructure. Operators tend to share their active equipment, such as OLTs. But the fiber infrastructure is generally not shared. You do not want an FTTH operator to unplug an antenna site

by mistake. Please rephrase.

Response:

We thank the reviewer for the suggestion in the description of infrastructure sharing. To address this concern, we have revised the relevant text on page 3 of the revised manuscript as follows: This convergence primarily facilitates the sharing of existing infrastructure such as optical line terminals (OLTs), resulting in substantial reductions in both capital and operational expenditures while facilitating streamlined service delivery [36,37].

Comment 4:

P6 fig 2: Is the “OLT” emitter using dual polarization? or single polarization as represented?

Response: We thank the reviewer for pointing out the mistake in the figure. We used a dual-polarization modulator at the OLT side. To clarify this, in the revised manuscript, we have labeled the component as “DP IQM.” in Fig. 2 (now Fig. 3) and added an explicit description “dual-polarization IQ modulator” to the caption of Figure 2 (now Figure 3).

Comment 5:

Page 7: the experimental setup should be described in more details: gain of the EDFA, wavelength of the LOs, linewidth of the LOs, insertion losses of the IQ MZM.

Response:

We thank the reviewer for this valuable suggestion. In this revised version, we have provided more details about the experimental set-up in the second paragraph of Section 3.1, as follows: After resampling to match the DAC sampling rate, the generated electrical waveforms are used to drive the DP-IQ modulator (Fujitsu FTM7992HM) with an insertion loss of 14 dB, which is biased at the null point to modulate the continuous-wave output from an external cavity laser (Ovlink, TSP-1000) with a typical linewidth of 25 kHz, centered at 1549.49 nm. The output power of the modulator is -10.4 dBm. Thus, the optical signal is then amplified to 10.6 dBm using an Erbium-doped fiber amplifier (EDFA, Amonics) operating with a gain of 21 dB, before being transmitted over a 20-km standard single-mode fiber link.

Comment 6:

Page 7: what is the frequency offset between emitters and LOs?

Response:

We thank the reviewer for this question. In our experiment, we employ conventional intradyne coherent detection for both ONUs and RRUs. Therefore, the wavelength of local oscillator is set to match that of the transmitter laser at 1549.49 nm. However, due to inherent frequency drift in the free-running lasers, a typical residual frequency offset of ± 200 MHz exists between the emitter and LOs. In the third paragraph of section 3.1 of the revised manuscript, we have revised the description of the coherent receiver, as follows: Conventional intradyne coherent detection is employed at both the ONU and RRU, with the local oscillator (LO) wavelength fixed at 1549.49 nm, identical to that of the transmitter. However, due to the inherent frequency drift of the lasers, a frequency offset of approximately 200 MHz exists between the transmitter and the LOs.

Comment 7:

P 11 fig 5: a few points are out of scale.

Response:

We thank the reviewer for pointing it out. We have now adjusted the y-axis limits in the figure 5 (now Figure 6) in this revised manuscript to fully accommodate all data points.

Comment 8:

P 11 fig 5: figure is difficult to read as the 2nd y-axis mixes EVM (%) and SNR (dB). Maybe the solution would be to propose a figure with BER, another with the four EVMs, and another with the 4 SNRs.

Response:

We thank the reviewer for this suggestion. To improve readability, we have restructured the original Fig. 5 (now Figure 6) into three separate subfigures: Figure 6(a) for BER versus ROP, Figure 6(b) for EVM versus ROP, and Figure 6(c) for SNR versus ROP.

Comment 9:

P 10: could the authors comment on the 109 km reach? usually, eCPRI is latency sensitive and thus limited to 20 km reach typically. Otherwise, latency is too high from

Fronthaul and low-layer splits.

Response:

We thank the reviewer for raising this important point regarding the latency requirements in mobile fronthaul applications. As the reviewer correctly noted, eCPRI-based mobile fronthaul employing low-layer functional splits is typically restricted to approximately 20 km due to stringent latency constraints. The 109 km field-deployed fiber in our experiment was chosen to demonstrate the feasibility of the proposed scheme for long-reach PON applications and access-metro network with low-latency analog radio-over-fiber [40,43,49], and more importantly, its robustness against real-world impairments over extended distances, including polarization fluctuations, and mechanical vibrations—that are difficult to fully replicate in a controlled laboratory environment. Nevertheless, the proposed scheme is highly flexible and can support the integrated deployment of short-reach PON, long-reach PON, and latency-sensitive mobile fronthaul, with fiber lengths tailored to the specific requirements of each use case, provided that sufficient optical power budget is available to maintain the required signal quality.

To address this comment, we have revised the description of the field trial in the first paragraph of Section 3.3 as follows: These results confirm its suitability for long-reach PON applications [59] and, more importantly, demonstrate its robustness against real-world impairments over extended distances, such as polarization fluctuations, and mechanical vibrations—that are difficult to fully replicate in a controlled laboratory environment. Note that, in practical deployments, the fiber length allocated to latency-sensitive mobile fronthaul can be limited to within 20 km while longer reaches can be reserved for PON services.

Comment 10:

P 13 fig 8 : similar comments than those to fig 5

Response:

We thank the reviewer for this comment. Similar to our revisions for Fig. 5, we have restructured the original Fig. 8 (now Figure 9 in revised manuscript) into three separate sub-figures to improve readability: Figure 9(a) for BER versus ROP, Figure 9(b) for EVM versus ROP, and Figure 9(c) for SNR versus ROP.

Comment 11:

Page 15: author affirm that modulations such as 65536 QAM can be supported, without

supporting that claim. The experiments go as high as 16384 QAM.

Response:

We thank the reviewer for this valuable comment. We have removed the statement “such as 65536-QAM or beyond” from the first paragraph of the Discussion section in the revised manuscript.

Comment 12:

Page 17: a reference is needed beside Carson’s rule.

Response:

We thank the reviewer for this suggestion. We have now added a reference to the seminal work on Carson’s rule [54].

[R3] Lathi, B.P., Ding, Z.: Modern Digital and Analog Communication Systems, 3rd edn. Oxford University Press, New York, USA (1998) (Ref. [54] in the revised manuscript)

Comment 13:

There is no proper “conclusion” section in the paper.

Response:

We thank the reviewer for this suggestion. In the revised manuscript, we have included a new Section 5 (“Conclusion”) to provide a clear summary of the key findings and future outlook.

Comment 14:

Typos: Bibliography: capital letters vanished in some references, such as in [11]. The “g” of “100g” should be capital. Please check all references.

Response:

We thank the reviewer for this careful proofreading. We have thoroughly checked the entire Reference list and corrected all capitalization inconsistencies and typos, including restoring the capital “G” in abbreviations such as “100G” throughout the bibliography.

Response to Reviewers 3 and 4

Comment:

The authors presented a transmission method for the future convergence of mobile and fixed optical access networks (e.g. PON), with the motivation that coherent technology is a potential candidate for next generation fixed access networks. They proposed to use coherent transmission to transmit the analog signal of the mobile channel within the phase of a digital unipolar PAM signal of the fixed access channel, using phase modulation (PM). Higher order QAM cardinalities in the mobile channel within the optical channel are enabled by an increased SNR, through a PM modulation index optimization. The proposed scheme is validated in a short reach point-to-point experimental setup, and in a metro point-to-point field demonstration, at 128 Gbps together with a 20 GHz aggregation bandwidth wireless signal. The concept is novel and can be relevant for next generation optical access communications. However, the experimental and field demonstrations were not performed in a point to multipoint fixed access network, like PON. In my opinion, the experimental demonstration and discussion as presented in the manuscript fails to answer the research question for the motivating application scenario. Also, the methods and experiment are presented for a downstream channel, meaning from the OLT/CU to the ONU and RRU, but it is not clear how the proposed method would work for the upstream channel, meaning from the ONU or RRU, to the OLT/CU. The paper is an extension of the ECOC2025 post-deadline session 03.04 (Reference [54]). From [54] and the post-deadline session presentation, it is not clear what additional information is introduced by the extended paper. I have the following additional observations

Response:

We sincerely thank the reviewers for the constructive feedback and positive assessment of the novelty and relevance of our work. We have carefully studied the comments raised by the reviewers and have made corrections accordingly. All changes made in the revised version are coloured in blue.

Regarding the point-to-multipoint nature of ONUs, the primary challenge arises from the power-splitting loss introduced by the optical splitter. In our experiments, we employed a variable optical attenuator to emulate different splitting ratios and to evaluate the optical power budget which is related to a function of the number of ONUs. This is a widely accepted method [R4 (Ref. [3] in the manuscript),R5,R6,R7] in PON research for assessing point-to-multipoint performance with adjusted splitting ratios. The approach effectively isolates the dominant effect of power division on received optical power sensitivity. For instance, a typical 20 km transmission distance in PON systems [10] results in approximately 4 dB of fiber attenuation (assuming 0.2 dB/km). The relationship between the number of ONUs and the required power budget can then be expressed as

Figure R1: Upstream based on time and wavelength division multiplexing schemes.

$$\text{Power budget (dB)} = \text{Fiber loss} + 3 \log_2 N, \quad (1)$$

where N is the number of ONUs. This accounts for the fiber loss plus the splitting loss (approximately 3 dB per doubling of the number of ONUs). As an example, supporting 256 ONUs would require a power budget of approximately 28 dB when considering only fiber and splitting losses. To address the reviewer’s comment, we have incorporated the following explanation into the third paragraph of Section 3.1: *At the receiver side, we employ a variable optical attenuator (VOA) to emulate various splitting ratios introduced by the optical splitter in a typical point-to-multipoint access network [3], thereby evaluating the received optical power (ROP) sensitivity and optical power budget.*

Regarding upstream transmission, we acknowledge that the proposed amplitude–phase layered modulation scheme is primarily designed for downstream transmission in integrated fixed-mobile access networks and remains fully compatible with existing upstream architectures, which typically use distinct wavelengths to suppress Rayleigh backscattering in bidirectional fiber transmission [10]. For next-generation fully integrated fixed–mobile upstream transmission, digital signals from ONUs and analog signals from RRUs can be allocated to distinct time slots and/or wavelengths, as shown in Fig. R1. This approach enables straightforward time- and wavelength-domain multiplexing [R8], allowing simultaneous reception of both signal types using a single coherent receiver at the OLT and thereby offering substantial potential for hardware simplification. Exploring such upstream integration techniques represents a promising and important direction for future research. To address this comment, we have incorporated the following discussion into the revised Discussion section: *The proposed amplitude–phase layered modulation scheme is primarily designed for downstream transmission in integrated fixed-mobile access networks. It remains fully compatible with existing upstream architectures, where conventional access networks typically employ distinct wavelengths to mitigate Rayleigh backscattering in bidirectional fiber transmission [10]. Looking forward to next-generation fully converged fixed–mobile upstream transmission, digital signals from*

ONUs for fixed access and analog signals from RRUs for mobile fronthaul can be multiplexed either in different time slots on the same wavelength, or in the same time slot using adjacent wavelengths. These time- and wavelength-domain multiplexing schemes [60] enable simultaneous reception of both signal types with a single coherent receiver at the OLT, offering a promising path toward reduced hardware complexity. Investigating such upstream integration strategies constitutes an important and exciting direction for future research.

Finally, this manuscript significantly extends our ECOC 2025 post-deadline paper by introducing a comprehensive fixed-mobile converged network architecture (Section 2.1), providing detailed theoretical principles (Sections 2.2 and 2.3), presenting additional experimental results (Section 3: Figs. 4, 5, 6, 8), and offering an in-depth discussion (Table 1 and Section 4: Discussion) of performance trade-offs and practical implications. We have added the following clarification in the last paragraph of Introduction: This work extends our post-deadline paper presented at ECOC 2025 [55]. It introduces a comprehensive fixed-mobile converged network architecture, provides a detailed elucidation of the underlying principles, presents substantial additional experimental results, and includes an in-depth discussion encompassing thorough performance comparisons, compatibility with existing systems, low-cost implementation alternatives, and considerations for upstream transmission.

Thank you again for the valuable suggestions, which have substantially strengthened the manuscript.

[R4] Erkılınc, M., Lavery, D., Shi, K., Thomsen, B., Killey, R., Savory, S., Bayvel, P.: Bidirectional wavelength-division multiplexing transmission over installed fibre using a simplified optical coherent access transceiver. *Nature communications*, vol. 8, no. 1, 1043 (2017) (Ref. [3] in the manuscript)

[R5] D. Huang *et al.*, “200 Gbps flexible coherent PD-NOMA PON in uplink and downlink with >35-dB power budget using successive interference cancellation,” *Opt. Fiber Technol.*, vol. 95, p. 104466, Jan. 2025.

[R6] T. Kanai, M. Fujiwara, R. Igarashi, N. Iiyama, R. Koma, J. Kani, and T. Yoshida, “Symmetric 10 Gbit/s 40-km reach DSP-based TDM-PON with a power budget over 50 dB,” *Opt. Express*, vol. 29, no. 11, pp. 17499–17509, May 2021, doi: 10.1364/OE.421917.

[R7] I. B. Kovacs, M. S. Faruk, and S. J. Savory, “A minimal coherent receiver for 200 Gb/s/ λ PON downstream with measured 29 dB power budget,” *IEEE Photon. Technol. Lett.*, vol. 35, no. 5, pp. 257–260, Mar. 2023.

[R8] Nakayama, Y., Hisano, D.: Wavelength and bandwidth allocation for mobile fronthaul in TWDM-PON. *IEEE Transactions on Communications* 67(11), 7642–7655 (2019) (Ref. [60] in the manuscript)

Comment 1:

The paper claims that digital transmission is improved by using the phase insensitive modulation unipolar PAM, but it is unclear what is being used as a basis for comparison to claim an improvement. Since the phase of the signal is used for the mobile signal transmission, there is fundamentally no gain in received optical power tolerance coming from the use of coherent detection. On the other hand, for a constant data rate, the use of unipolar PAM requires double the transmission bandwidth than the required by a QPSK modulation, therefore the simplification in the DSP from the use of a phase insensitive modulation is traded off with the requirement of a higher transmission bandwidth.

Response:

We thank the reviewer for this insightful comment and apologize for the lack of clarity. The baseline for comparison is bipolar PAM modulation with coherent detection. We have rephrase the sentences in the Abstract: This approach not only simplifies the coherent reception for fixed optical access by rendering it phase-insensitive and eliminating the need for carrier phase recovery in digital signal processing,.... and the second paragraph of Section 2.2: Compared with conventional bipolar PAM, the proposed phase-insensitive unipolar PAM modulation not only simplifies coherent DSP for the digital layer—eliminating the need for FOE and CPR—but also enables simultaneous phase modulation for analog fronthaul signals, thereby overcoming the inherent SNR ceiling in traditional analog radio-over-fiber links.

Regarding the bandwidth trade-off, we fully agree with the reviewer. For a fixed data rate, unipolar PAM requires approximately twice the electrical bandwidth compared to QPSK, as only one dimension (intensity) is utilized for the digital signal while the phase dimension carries the analog fronthaul waveform. This is indeed a key trade-off between DSP simplification, flexibility and bandwidth efficiency. To address this point explicitly, we have added the following clarification in the second paragraph of Discussion section: However, it should also be noted that the achievable data rate for the digital transmission is approximately halved compared to full IQ modulation on both polarizations. This represents a trade-off: simplified DSP complexity and enhanced flexibility for hybrid digital-analog transmission are achieved at the expense of reduced bandwidth efficiency.

Comment 2:

In page 3 the paper claims “In this work, we experimentally demonstrate an integrated fixed– mobile optical access network architecture in a field trial, enabled by the proposed amplitude– phase layered modulation.” But what is understood from the paper is that the field demonstration was performed in a metro network.

Response:

We thank the reviewer for this careful comment. The experimental demonstration was indeed conducted over both a 20-km laboratory fiber link (representing typical short-reach access scenarios) and a 109-km field-deployed fiber link in Hong Kong (to validate performance in long-reach PON applications, metro-access network [40,43,49] and more importantly robustness under realistic impairments). This field-deployed infrastructure forms part of the regional optical access network serving universities and data centers.

To avoid any potential ambiguity and more accurately reflect the experimental scope, we have rephrased the statement in the last paragraph of Introduction as follows: In this work, we experimentally demonstrate an integrated fixed–mobile optical access network architecture through laboratory experiments and transmission over field-deployed fibers, enabled by the proposed amplitude–phase layered modulation.

Comment 3:

In page 3 it is mentioned “However, conventional ARoF is limited by an signal-to-noise ratio (SNR) ceiling of approximately 25 dB [28], imposed by fiber channel impairments such as transceiver nonlinearities and noise, . . .”, however transceivers nonlinearities and noise are not channel effects coming from the fiber itself. I would recommend rephrasing it.

Response:

We thank the reviewer for the comment on improving the technical accuracy of the manuscript. We have now rephrased the sentence in the third paragraph of Introduction, as follows: However, conventional ARoF is limited by a signal-to-noise ratio (SNR) ceiling of approximately 25 dB [30], imposed by system impairments such as transceiver nonlinearities and electrical noise.

Comment 4:

In page 3 “. . . enabled by the proposed amplitude–phase layered modulation. This scheme allows for simplified digital coherent optics without the need for FOE and CPR, . . .”. In my opinion, what is simplified is the coherent DSP.

Response:

We thank the reviewer for this precise observation. We agree that the simplification is done for the coherent DSP. We have rephrased the sentence in the last paragraph of Introduction in the revised manuscript as follows: This scheme allows for simplified

coherent DSP by eliminating the need for FOE and CPR.

Comment 5:

In fact, for the digital channel, the coherent optics become more complex, since the proposed scheme requires double bandwidth when compared with coherent modulation, and the use of coherent transmitters and receivers, when compared with IM/DD.

Response:

We thank the reviewers' valuable comments. Indeed, for the same data rate in the digital transmission, unipolar PAM requires approximately twice the electrical bandwidth compared to dual-polarization IQ modulation (e.g., QPSK), as it utilizes only intensities of both polarizations for digital modulation while uses phase for analog modulation. Conversely, with the same electrical bandwidth, the data rate of the unipolar PAM layer is roughly half that of a full coherent modulation system but the analog layer enables the transmission of additional analog signals. This is indeed a key trade-off between DSP simplification, flexibility and bandwidth efficiency. To address this point explicitly, we have added the following clarification in the Discussion section: *However, it should also be noted that the achievable data rate for the digital transmission is approximately halved compared to full IQ modulation on both polarizations. This represents a trade-off: simplified DSP complexity and enhanced flexibility for hybrid digital-analog transmission are achieved at the expense of reduced bandwidth efficiency.*

The proposed scheme can also be implemented using simple IM-DD systems by employing digital subcarrier multiplexing (DSCM). In this approach, the converged signal is digitally up-converted to an intermediate frequency (IF), and only the real part of the shifted signal is used to drive the intensity modulator. In this way, only intensity modulation is required at the transmitter. Furthermore, the receiver can be further simplified to a single photodetector for envelope detection. A digital hybrid is then employed to down-convert the IF signal to baseband, thereby further reducing hardware complexity and cost in this IM-DD implementation. We added this point in the Discussion: *To address cost and power consumption concerns, the proposed scheme can also be implemented in IM-DD systems, making it compatible with current access networks that predominantly employ cost-effective optics such as directly-modulated lasers and electro-absorption modulated lasers. In this alternative approach, digital subcarrier multiplexing is used: the converged complex-valued baseband signal is digitally frequency-shifted to an intermediate frequency (IF) [28], and only the real part of the shifted signal is taken to drive the intensity modulator. In this way, only intensity modulation is required at the transmitter. Furthermore, the receiver can be further simplified to a single photodetector*

for envelope detection. A digital hybrid is then employed to down-convert the IF signal to baseband, thereby further reducing hardware complexity and cost in this IM-DD implementation.

Comment 6:

The experimental setup is not completely represented in figure number 2. The figure should represent an accurate complete description of the experimental method performed, that simultaneously shall be completely described in the text.

Response:

We thank the reviewer for this valuable suggestion. To provide a more accurate and complete representation of the experimental setup, we have revised the original Fig. 2 (now Fig. 3 in the revised manuscript) with additional details, including the dual-polarization IQ modulator, laser wavelengths, launch power, and fiber length. These parameters are also explicitly described in the accompanying text in the second paragraph of Section 3.1.

Comment 7:

In page 7, the bandwidth of the IQ modulator is not mentioned.

Response:

We thank the reviewer for the valuable comment. We have added the key parameters and model number of the IQ modulator in the second paragraph of Section 3.1 in the revised manuscript as follows: dual-polarization IQ modulator (Fujitsu FTM7992HM) with a typical insertion loss of 14 dB and a 3-dB bandwidth of 35 GHz. In addition, key parameters of the lasers and EDFAs are also provided in the revised manuscript.

Comment 8:

In page 7, it is mentioned that EDFAs boost the transmitted optical power to 10.6 dBm. This power value directly affects the obtained results on the received optical power (ROP) sensitivity and optical power budget. I would like the authors to comment on why they decided this value and how it relates to the considered fixed access optical network application.

Response:

We thank the reviewer for this comment. A higher launch power can potentially increase the power budget for the fixed access optical network. In this experiment, the launch power of 10.6 dBm was optimized to balance ROP sensitivity and avoidance of fiber nonlinearities (primarily self-phase modulation). We have added the following clarification at the end of the second paragraph of Section 3.1: The launch power of 10.6 dBm is selected through experimental optimization. Higher launch powers would induce fiber nonlinearities—primarily self-phase modulation—leading to degradation in system performance.

Comment 9:

In page 7 it is mentioned, “First, we focus on digital transmission at the ONU side.”. However, a digital reception at the ONU side and transmission at the OLT side is presented.

Response:

We thank the reviewer for pointing out this mistake. We have revised the sentence in the second paragraph of Section 3.2 in the revised manuscript as follows: First, we focus on the reception of the unipolar PAM-modulated digital signal at the ONU side.

Comment 10:

In Fig 4 (a) we observe the SNR vs modulation index for a 1-GHz 1024-QAM phase modulated signal. The text doesn't mention that Fig 4 (a) is for a 1-GHz 1024-QAM and is only mentioned in the figure caption.

Response:

We thank the reviewer for this careful observation. We have added an explicit description in the third paragraph of Section 3.2, as follows: To this end, the modulation index is swept to assess the achievable SNR as a function of modulation index, as illustrated in Fig. 5(a) for a 1-GHz 1024-QAM phase-modulated signal.

Comment 11:

I think it would be interesting to know if and how the behavior presented in Fig. 4 (a) changes if the SNR vs modulation index is made with a different bandwidth or a different M-QAM constellation cardinality.

Figure R2:SNR versus QAM Order

Response:

The trends observed in Fig. 4(a) (now Fig. 5(a) in the revised manuscript) are largely robust to changes in M-QAM constellation cardinality. As shown in Figure R2, we simulated the SNR as a function of QAM order under AWGN channel with a fixed modulation index. The variations in SNR across different QAM orders are negligible. This robustness arises because the resulting analog waveforms, after the IFFT in OFDM generation, exhibit similar statistical characteristics—approaching a Gaussian distribution—for different high-order QAM formats.

In contrast, increasing the signal bandwidth reduces the achievable SNR due to the transceiver’s limited bandwidth and the elevated noise floor at higher frequencies, necessitating a higher modulation index to maintain performance. In our experiments, the modulation indices were optimized accordingly to meet the EVM thresholds (now indicated by dashed lines in the revised Figure 5) while providing sufficient margin to accommodate increased aggregation bandwidth.

To address the reviewer’s comment, we have added the following clarification in the revised manuscript: *Accordingly, the modulation indices are experimentally optimized to 11, 12, 22, and 44 for 256-QAM, 1024-QAM, 4096-QAM, and 16384-QAM, respectively. These values ensure that the EVM remains below the corresponding 3GPP-specified thresholds while providing sufficient margin to accommodate increased aggregation bandwidth. The additional margin is necessary because wider signal bandwidths degrade the SNR at higher frequencies due to the transceiver’s limited bandwidth and elevated noise floor.*

Comment 12:

Additionally, from the text in page 10 “Based on the 3GPP technical specifications and the derivation in [29, 56], we then determine the appropriate modulation indices for different QAM formats. Specifically, the EVM (SNR) requirements are 3.5% (29.1 dB) for 256-QAM, 2.5% (32 dB) for 1024-QAM, 1.29% (37.8 dB) for 4096-QAM, and 0.66% (43.6 dB) for 16384-QAM. Accordingly, we set the modulation index to 11, 12, 22, and 44 for 256-QAM, 1024-QAM, 4096-QAM, and 16384-QAM, respectively, ensuring that the EVM remains below the specified threshold.”. It is unclear from the text how the modulation indexes are selected. E.g. are you using the theoretical 6.02 dB or with one or several SNR vs modulation index analyses?

Response:

We thank the reviewer for this valuable comment on the selection of modulation indices. The modulation indices were chosen experimentally based on measured EVM versus modulation index curves (as shown in Fig. 5(a) in the revised manuscript), where we have now added horizontal dashed lines indicating the 3GPP-specified EVM thresholds for 256-QAM to 16384-QAM. For each QAM format, the modulation index was selected to ensure the EVM remains below the corresponding threshold while providing sufficient margin to support increased aggregation bandwidths (as explored in Figs. 5(b)-(e)). This empirical approach accounts for practical transceiver impairments and ensures robust performance across the targeted bandwidth range, rather than relying solely on theoretical derivations. To improve clarity, we have revised the relevant text in the third paragraph of Section 3.2 as follows: Accordingly, the modulation indices are experimentally optimized to 11, 12, 22, and 44 for 256-QAM, 1024-QAM, 4096-QAM, and 16384-QAM, respectively. These values ensure that the EVM remains below the corresponding 3GPP-specified thresholds while providing sufficient margin to accommodate increased aggregation bandwidth. The additional margin is necessary because wider signal bandwidths degrade the SNR at higher frequencies due to the transceiver’s limited bandwidth and elevated noise floor.

Comment 13:

In the field demonstration and the setup presented in figure 6, it is not clear if optical amplifiers (e.g. EDFAs) are present within the optical link and how this affects the obtained optical power budget.

Response:

We thank the reviewer for this valuable comment. In the field demonstration, which was designed to validate the suitability for long-reach PON applications [59], metro-access network [40,43,49], and, more importantly, demonstrate its robustness against real-world impairments over extended distances, such as polarization fluctuations, and mechanical

vibrations, a single EDFA with 20 dB gain was inserted at an intermediate node, and a VOA was placed at the receiver side. These components were employed solely as experimental tools to evaluate the optical power budget under practical deployment conditions. Specifically, the EDFA compensated only for the excess splice and insertion losses along the 109 km field-deployed fiber route, where the total loss reached 41.4 dB—substantially higher than the expected 21.8 dB for standard single-mode fiber at 0.2 dB/km. In typical laboratory experiments or cleaner PON deployments with fewer splices, such an EDFA would not be required. Importantly, the incorporation of these components does not alter the intrinsic optical power budget characteristics of the proposed modulation scheme, as the optical power budget is fundamentally defined by the difference between the transmitted launch power and the required ROP at the receiver. For improved clarity, we have revised Figure 6 (now Figure 7) to explicitly indicate the EDFA (for experimental measurement) location and have updated the text in the second paragraph of Section 3.3 as follows: In this field demonstration, conducted as prototype validation of the proposed scheme, a 20-dB gain EDFA is inserted at an intermediate node and a VOA is placed at the receiver side. These components serve as experimental tools to facilitate parameter sweeping and accurate measurement of the optical power budget. The EDFA, employed solely for experimental measurement purposes, compensates only for the excess splice and insertion losses in the deployed fiber link and does not alter the intrinsic optical power budget characteristics of the proposed modulation scheme itself. In conventional PON deployments with cleaner fiber infrastructure and fewer splices, such an EDFA would not be required. This setup enables thorough validation of the scheme’s robustness against field-induced polarization variations and phase fluctuations.

Comment 14:

In figure 6 at the bottom a subfigure shows the optical link. The meaning from the icon that is depicted is unclear and shall be mentioned in the figure and the text (e.g. is it an optical switch? How does this affect the field demonstration?). The experimental setup used for the field demonstration needs to be fully described in the paper.

Response:

We thank the reviewer for this valuable comment. The icon in the original figure was intended to represent intermediate link nodes (e.g., patch panels or access points) along the field-deployed route, which introduce additional splice and insertion losses. To eliminate any ambiguity, we have removed the icon and revised the original Fig. 6 (now Fig. 7 in the revised manuscript) to provide a clearer illustration of the field-deployed optical link, with explicit labeling of the fiber segments, locations, and the EDFA placement for

experimental measurement. The complete experimental setup for the field demonstration, including transceiver configuration (identical to Section 3.1), fiber segment details, and EDFA insertion, is now fully provided in Fig. 6 (now Fig. 7) and described in the second paragraph of Section 3.3: The configurations are identical to those used in the previous setup (see Section 3.1) and thus we focus primarily on the field-deployed fiber link here. This link consists of four segments of SMF, totaling approximately 109 km: a 20 km segment (7 dB loss) from Hong Kong Polytechnic University (PolyU) to a data center in Tseung Kwan O; a 28.7 km segment (14 dB loss) from Tseung Kwan O to City University of Hong Kong (CityU); a 33.2 km segment (12 dB loss) from CityU to a data center in Chai Wan; and a 26.9 km segment (8.4 dB loss) from Chai Wan back to PolyU. The routes of these fiber segments are illustrated in Fig. 7(a). The total link loss of 41.4 dB substantially exceeds the expected value for standard SMF with a loss of 0.2 dB/km, primarily due to numerous splice losses accumulated along the field-deployed route. In this field demonstration, conducted as prototype validation of the proposed scheme, a 20-dB gain EDFA is inserted at an intermediate node and a VOA is placed at the receiver side. These components serve as experimental tools to facilitate parameter sweeping and accurate measurement of the optical power budget. The EDFA, employed solely for experimental measurement purposes, compensates only for the excess splice and insertion losses in the deployed fiber link and does not alter the intrinsic optical power budget characteristics of the proposed modulation scheme itself. In conventional PON deployments with cleaner fiber infrastructure and fewer splices, such an EDFA would not be required. This setup enables thorough validation of the scheme’s robustness against field-induced polarization variations and phase fluctuations.

Comment 15:

In Fig. 5, BER vs ROP of the digital channel is presented for different analog waveform channels. From the results of Fig. 3 e), it was claimed that the phase modulation doesn’t considerably affects the digital channel, but in Fig. 5 a) to d) the BER vs ROP achieved optical power budget is different. Same for Fig. 8 a) to e).

Response:

We thank the reviewer for this careful observation. The BER versus ROP curves for the digital channel under different phase-modulated analog waveforms (Figs. 5(a)–(d) and Figs. 8(a)–(e)) indeed show minor variations in achieved optical power budget. However, these differences are small and fall within the range of experimental fluctuations caused by random noise in the transmission system and repeatability of the measurement. Overall, the performance remains comparable across conditions, supporting our claim

that the phase-modulated layer has a negligible impact on the digital intensity-modulated layer. To clarify this point and avoid potential confusion, we have added the following note in the last paragraph of Section 3.2: The minor variations in optical power budget across different analog waveforms are attributable to experimental noise fluctuations and measurement repeatability, confirming the negligible crosstalk from the phase-modulated signal.

Comment 16:

Code availability. The authors publicly deposited the presumably MATLAB scripts APON_PFH_Transmitter.m and APON_PFH_Receiver.m in a Zenodo.org database. I couldn't run the supporting scripts, the database is missing, at least, the DMT function. I would recommend the authors review if all necessary functions are available in the database and to add a README file clarifying particular aspects of the scripts, like what software and software version is used to run them.

Response:

We thank the reviewer for this important feedback and apologize for the incomplete code deposit in the original submission. To address this issue, we have updated the Zenodo repository as follows:

1. Replaced the previous separate transmitter/receiver scripts with a single comprehensive Main.m script that integrates the full simulation flow.
2. Added all required custom functions and classes (including the OFDM/DMT implementation and supporting utilities) to ensure the code runs out-of-the-box.
3. Additionally, we have included a detailed README file that specifies: Required software: MATLAB R2024a. Necessary toolboxes: Communications Toolbox (recommended for scatterplot and related functions). Instructions for running the script and a brief overview of key configurable parameters.

Thank you for this valuable suggestion, which has significantly improved the usability of the deposited code.

Comment 17:

Data availability. The authors publicly deposited in Zenodo.org a database including the data from figures 3 to 10, in XLSX format and a folder with some constellation figures. I would recommend the authors add a short supplementary document indicating what information is contained in each of the files from the database, since this information

cannot be directly inferred from the file names and folder structures.

Response:

We thank the reviewer for this helpful recommendation. To enhance the usability and transparency of the deposited data, we have reorganized the Zenodo repository for better clarity and added a detailed README.txt file. This supplementary document allows users to quickly understand the content without relying solely on file names. Thank you for this suggestion, which significantly improves the reproducibility of our results.

Response to Reviewer 5

Comment:

This is a well written review paper that is an extension of the authors' previous ECOC post-deadline paper in which they proposed a digital/analog modulation scheme to support fixed/mobile optical fronthaul access. My comments mainly concern clarification of their results.

Response:

We sincerely thank the reviewer for the highly positive assessment of our work. We have carefully studied the comments raised by the reviewer and have made revisions accordingly. All changes made in the revised version are coloured in blue.

Comment 1:

The phase-modulated system uses OFDM, which has parameters such as number of carriers, guard interval, etc. The authors should include details of the OFDM implementation.

Response:

We thank the reviewer for this pertinent comment. To provide a clearer description of the OFDM implementation in the phase-modulated signal, we have added the following details in the second paragraph of Section 3.1: The OFDM signal is generated using a 1024-point inverse fast Fourier transform (IFFT). Among the 1024 subcarriers, 900 are employed as effective data-carrying subcarriers, while the remaining subcarriers are zero-padded at the spectrum edges to prevent aliasing. A cyclic prefix of 8 samples is appended to each OFDM symbol to mitigate inter-symbol interference arising from channel dispersion. Note that key parameters, including the IFFT size, number of active subcarriers, and cyclic prefix length, can be adjusted depending on the specific experimental requirements and bandwidth configuration [56].

Comment 2:

In Figs. 5 and 8, the authors opted to evaluate performance using BER for the amplitude-modulated digital system, but EVM for the phase-modulated "analog" system which actually represents digital mobile data. Why the difference? BER should be a superior metric for system performance, as BER can only be accurately inferred by EVM when noise is Gaussian, which is not guaranteed or proven in the paper. Since the authors generated phase-modulated OFDM signals containing digital data, it should not be difficult to just

estimate BER? At the very least, the users should justify the use of EVM rather than BER.

Response:

We thank the reviewer for this insightful comment. The phase-modulated layer conveys analog fronthaul waveforms from the antennas, treated as continuous-time analog signals in the radio-over-fiber transmission framework. Accordingly, relevant industry standards—such as 3GPP TS 38.101-2 (Release 16) [56]—primarily specify signal quality in terms of error vector magnitude (EVM) to assess waveform fidelity, rather than bit error rate (BER) of the underlying digital data. To clarify this rationale and address potential confusion, we have revised the manuscript as follows: The amplitude-modulated digital signal is evaluated using bit error rate (BER). For the phase-modulated layer carrying analog fronthaul signals, transmission fidelity is assessed via SNR and error vector magnitude (EVM) [19,28,42], where EVM (linear) \$\approx 1/\sqrt{\text{SNR}}\$. This aligns with relevant industry standards (e.g., 3GPP TS 38.101-2 [56]), which specify EVM as the primary quality metric for wireless signals.

[R8] Zhang, C., Zhu, Y., He, B., Lin, J., Liu, R., Xu, Y., Yi, L., Zhuge, Q., Hu, W., Hu, W., et al.: Clone-comb-enabled high-capacity digital-analogue fronthaul with high-order modulation formats. *Nature Photonics* 17(11), 1000–1008 (2023) (Ref. [19] in the revised manuscript)

[R9] Ishimura, S., Takahashi, H., Tsuritani, T., Suzuki, M.: SNR enhancement of up to 9.5 dB utilizing four-wave mixing for angle-modulated analog optical links. *Journal of Lightwave Technology* 40(5), 1464–1471 (2021) (Ref. [28] in the revised manuscript)

[R10] Dass, D., Delmade, A., Barry, L., Roeloffzen, C.G., Geuzebroek, D., Browning, C.: Wavelength & mm-wave flexible converged optical fronthaul with a low noise Si-based integrated dual laser source. *Journal of Lightwave Technology* 40(10), 3307–3315 (2022) (Ref. [42] in the revised manuscript)

[R11] 3GPP: NR; User Equipment (UE) radio transmission and reception; Part 2: Range 2 Standalone (TS 38.101-2 version 16.19.0 Release 16). (Ref. [56] in the revised manuscript)

Comment 3:

In Figs. 4 and 7, the authors first performed a sweep of EVM/SNR versus modulation index to find the k_p necessary to support various QAM formats, and then swept EVM/SNR against the aggregation bandwidth of the OFDM signal to find the maximum bandwidth that can be supported. The thresholds for each modulation format is a bit hard to see. Maybe the authors can add dotted lines for the threshold EVM (if this is

even an acceptable metric, or else BER can be used) so we can see where the blue lines cross the threshold.

Also, the two vertical axes may be redundant. I believe they show the same information as $EVM = 1/\text{SNR}$. If this is correct, the authors should state this relationship in the paper.

Response:

We thank the reviewer for these helpful and practical suggestions. We have added horizontal dashed lines in the revised Figs. 5 and 8 (previously Figs. 4 and 7) to clearly indicate the maximum EVM thresholds required for reliable operation of each QAM format, making it easier to identify the achievable modulation index and aggregation bandwidth. We confirm that, under AWGN-dominated conditions, EVM (linear) $\approx 1/\sqrt{\text{SNR}}$. We have retained both axes for convenience, as EVM is the standard metric specified in 3GPP requirements while SNR provides intuitive validation against theoretical expectations.

To improve clarity, we have explicitly stated this relationship in last paragraph of Section 3.1: *The amplitude-modulated digital signal is evaluated using bit error rate (BER). For the phase-modulated layer carrying analog fronthaul signals, transmission fidelity is assessed via SNR and error vector magnitude (EVM) [19,28,42], where EVM (linear) $\approx 1/\sqrt{\text{SNR}}$. This aligns with relevant industry standards (e.g., 3GPP TS 38.101-2 [56]), which specify EVM as the primary quality metric for wireless signals.*

[R11] 3GPP: NR; User Equipment (UE) radio transmission and reception; Part 2: Range 2 Standalone (TS 38.101-2 version 16.19.0 Release 16). (Ref. [56] in the revised manuscript)

Comment 4:

Likewise, it would help the reader to show dashed lines for the BER threshold and minimum ROP in Figs. 3(e-g), 5(a-d) and 8(a-d) for results of amplitude-modulated system.

Response:

We thank the reviewer for this helpful suggestion. We have added horizontal dashed lines indicating the 7% overhead HD-FEC BER threshold for the digital (amplitude-modulated) signal and the corresponding EVM thresholds for the analog (phase-modulated) signal in Figs. 3, 5, and 8 (now Figs. 4, 6, and 9) in the revised manuscript.

Comment 5:

Figs. 5 and 8, which combine results for both the amplitude-modulated and phase-modulated systems, seems cluttered. The right vertical axes are also confusing as they show EVM (%) and SNR (dB) on the same scale, but they are completely different things. If EVM and SNR are inverses of each other, maybe you don't need to show both. Furthermore, if the phase-modulated system was evaluated using BER rather than EVM/SNR, the two sets of BERs can be shown on one vertical axis and would be easier to read.

I observe saturation in the EVM/SNR of the phase-modulated system. I assume this is due to transceiver SNR. This should be discussed and stated in the paper.

Response:

We thank the reviewer for these valuable suggestions, which have significantly improved the clarity of the results presentation. To address the clutter and dual-axis confusion in the original Figs. 5 and 8 (now, Figs. 6 and 9), we have reorganized the data into three separate figures in the revised manuscript: 1. One figure showing BER of digital bits versus ROP when carrying different modulation formats of analog signals, 2. One showing SNR versus ROP for different modulation formats of analog signals, 3. One showing EVM versus ROP for different modulation formats of analog signals. This separation eliminates the potential confusion and redundant axes while allowing direct comparison where needed. Additionally, we have clarified the relationship between EVM and SNR, $EVM(\text{linear}) \approx 1/\sqrt{SNR}$, in the text (See Comment 3).

Regarding the observed saturation in EVM/SNR for the phase-modulated layer at high ROP, this is indeed limited by the finite electrical SNR of the transceiver hardware. We have added the following discussion in the last paragraph of Section 3.2: The saturation in EVM and SNR observed at higher ROPs is attributed to the inherent SNR floor of the transceiver components.

Comment 6:

On page 14, "...bandwidths ranging from 36 GHz (64-QAM) to 4 GHz (16384-QAM)". It should be noted that these number are a summation of the bandwidths of both polarizations.

Response:

We thank the reviewer for this careful observation. To avoid potential ambiguity, we have clarified in the Section 5 (Conclusion) that the reported bandwidths represent the aggregate values across both polarizations. The sentence now reads:...dual-polarization aggregated bandwidths ranging from 36 GHz (for 64-QAM) to 4 GHz (for 16384-QAM).

Comment 7:

In the introduction and in Fig. 8(e), the authors cited a number of 61.4 bit/s to represent 1 Hz of wireless data. Where does this number come from? I cannot find this number in the cited references. CPRI samples and digitizes the analog waveforms of the wireless signal (with additional overhead). The ratio between the required CPRI data rate vs wireless data rate should not be a constant. Furthermore, if the aggregation bandwidth was X, then in case CPRI was used, the achievable wireless data rate Y would be much lower than X due to a low spectral efficiency. The correct comparison should be the data rate achieved ($X \text{ Gbaud} * B \text{ bits/symbol}$) with the proposed method versus data rate Y which would be achieved using CPRI.

Response:

We thank the reviewer for this detailed and important comment regarding the origin and interpretation of the 61.4 bit/s/Hz used for CPRI-equivalent rate.

This value originates from calculations in Section IV.B of reference [7], the first paragraph of the Introduction in [19], and Eq. (1) in [20]. It represents a typical overhead factor for CPRI in a common LTE configuration (20 MHz carrier bandwidth with 2×2 MIMO), yielding a line rate of approximately 2.4576 Gbit/s [7] and an effective ratio of about 61.4 bit/s/Hz ($= 2.4576 \text{ Gbit/s} / (20 \text{ MHz} \times 2 \text{ antennas})$) of wireless spectrum. This figure has been widely adopted in pioneering works on analog fronthaul [13, 19, 25, 31, 32].

We fully agree that the ratio is configuration-dependent, varying with parameters such as carrier bandwidth, MIMO layers, quantization bits, control overhead, and line coding. The CPRI line rate is given by $\text{Data rate} = M \times S_r \times N \times 2(I/Q) \times C_w \times C$, where M is the number of antennas, S_r the sampling rate, N the sample width (typically 15 bits), C_w the control word factor (16/15), and C the coding overhead (e.g., 10/8 for 8B/10B).

To prevent misunderstanding and better reflect this variability, we have revised the relevant description by rephrasing it and adding references in the Introduction as follows: However, each gigahertz of wireless bandwidth typically requires approximately 61.4 Gb/s of fronthaul capacity when using digitized CPRI in representative mobile network configurations, resulting in very low wireless-to-optical bandwidth efficiency [7,19,20].

[R12] Pizzinat, A., Chanclou, P., Saliou, F., Diallo, T.: Things you should know about fronthaul. *Journal of Lightwave Technology* 33(5), 1077–1083 (2015) (Ref. [7] in the revised manuscript)

[R13] Zhang, C., Zhu, Y., He, B., Lin, J., Liu, R., Xu, Y., Yi, L., Zhuge, Q., Hu, W., Hu, W., et al.: Clone-comb-enabled high-capacity digital-analogue fronthaul with high-order modulation formats. *Nature Photonics* 17(11), 1000–1008 (2023) (Ref. [19] in the revised manuscript)

[R14] Che, D.: Analog vs digital radio-over-fiber: A spectral efficiency debate from the SNR perspective. *Journal of Lightwave Technology* 39(16), 5325–5335 (2021) (Ref.

[20] in the revised manuscript)

The authors presented a transmission method for the future convergence of mobile and fixed optical access networks (e.g. PON), with the motivation that coherent technology is a potential candidate for next generation fixed access networks. They proposed to use coherent transmission to transmit the analog signal of the mobile channel within the phase of a digital unipolar PAM signal of the fixed access channel, using phase modulation (PM). Higher order QAM cardinalities in the mobile channel within the optical channel are enabled by an increased SNR, through a PM modulation index optimization. The proposed scheme is validated in a short reach point-to-point experimental setup, and in a metro point-to-point field demonstration, at 128 Gbps together with a 20 GHz aggregation bandwidth wireless signal. The concept is novel and can be relevant for next generation optical access communications. However, the experimental and field demonstrations were not performed in a point to multipoint fixed access network, like PON. In my opinion, the experimental demonstration and discussion as presented in the manuscript fails to answer the research question for the motivating application scenario. Also, the methods and experiment are presented for a downstream channel, meaning from the OLT/CU to the ONU and RRU, but it is not clear how the proposed method would work for the upstream channel, meaning from the ONU or RRU, to the OLT/CU. The paper is an extension of the ECOC2025 post-deadline session 03.04 (Reference [54]). From [54] and the post-deadline session presentation, it is not clear what additional information is introduced by the extended paper. I have the following additional observations:

- The paper claims that digital transmission is improved by using the phase insensitive modulation unipolar PAM, but it is unclear what is being used as a basis for comparison to claim an improvement. Since the phase of the signal is used for the mobile signal transmission, there is fundamentally no gain in received optical power tolerance coming from the use of coherent detection. On the other hand, for a constant data rate, the use of unipolar PAM requires double the transmission bandwidth than the required by a QPSK modulation, therefore the simplification in the DSP from the use of a phase insensitive modulation is traded off with the requirement of a higher transmission bandwidth.
- In page 3 the paper claims "*In this work, we experimentally demonstrate an integrated fixed-mobile optical access network architecture in a field trial, enabled by the proposed amplitude-phase layered modulation.*" But what is understood from the paper is that the field demonstration was performed in a metro network.
- In page 3 it is mentioned "*However, conventional AroF is limited by an signal-to-noise ratio (SNR) ceiling of approximately 25 dB [28], imposed by fiber channel impairments such as transceiver nonlinearities and noise, ...*", however transceivers nonlinearities and noise are not channel effects coming from the fiber itself. I would recommend rephrasing it.
- In page 3 "... *enabled by the proposed amplitude-phase layered modulation. This scheme allows for simplified digital coherent optics without the need for FOE and CPR, ...*". In my opinion, what is simplified is the coherent DSP.
- In fact, for the digital channel, the coherent optics become more complex, since the proposed scheme requires double bandwidth when compared with coherent modulation, and the use of coherent transmitters and receivers, when compared with IM/DD.

- The experimental setup is not completely represented in figure number 2. The figure should represent an accurate complete description of the experimental method performed, that simultaneously shall be completely described in the text.
- In page 7, the bandwidth of the IQ modulator is not mentioned.
- In page 7, it is mentioned that EDFAs boost the transmitted optical power to 10.6 dBm. This power value directly affects the obtained results on the received optical power (ROP) sensitivity and optical power budget. I would like the authors to comment on why they decided this value and how it relates to the considered fixed access optical network application.
- In page 7 it is mentioned, "*First, we focus on digital transmission at the ONU side.*". However, a digital reception at the ONU side and transmission at the OLT side is presented.
- In Fig 4 (a) we observe the SNR vs modulation index for a 1-GHz 1024-QAM phase modulated signal. The text doesn't mention that Fig 4 (a) is for a 1-GHz 1024-QAM and is only mentioned in the figure caption.
- I think it would be interesting to know if and how the behavior presented in Fig. 4 (a) changes if the SNR vs modulation index is made with a different bandwidth or a different M-QAM constellation cardinality.
- Additionally, from the text in page 10 "*Based on the 3GPP technical specifications and the derivation in [29, 56], we then determine the appropriate modulation indices for different QAM formats. Specifically, the EVM (SNR) requirements are 3.5% (29.1 dB) for 256-QAM, 2.5% (32 dB) for 1024-QAM, 1.29% (37.8 dB) for 4096-QAM, and 0.66% (43.6 dB) for 16384-QAM. Accordingly, we set the modulation index to 11, 12, 22, and 44 for 256-QAM, 1024-QAM, 4096-QAM, and 16384-QAM, respectively, ensuring that the EVM remains below the specified threshold.*". It is unclear from the text how the modulation indexes are selected. E.g. are you using the theoretical 6.02 dB or with one or several SNR vs modulation index analyses?
- In the field demonstration and the setup presented in figure 6, it is not clear if optical amplifiers (e.g. EDFAs) are present within the optical link and how this affects the obtained optical power budget.
- In figure 6 at the bottom a subfigure shows the optical link. The meaning from the icon that is depicted is unclear and shall be mentioned in the figure and the text (e.g. is it an optical switch? How does this affect the field demonstration?). The experimental setup used for the field demonstration needs to be fully described in the paper.
- In Fig. 5, BER vs ROP of the digital channel is presented for different analog waveform channels. From the results of Fig. 3 e), it was claimed that the phase modulation doesn't considerably affect the digital channel, but in Fig. 5 a) to d) the BER vs ROP achieved optical power budget is different. Same for Fig. 8 a) to e).

Code availability

The authors publicly deposited the presumably MATLAB scripts *APON_PFH_Transmitter.m* and *APON_PFH_Receiver.m* in a Zenodo.org database. I couldn't run the supporting scripts, the database is missing, at least, the DMT function. I would recommend the authors review if all necessary functions are available in the database and to add a README file clarifying particular aspects of the scripts, like what software and software version is used to run them.

Data availability

The authors publicly deposited in Zenodo.org a database including the data from figures 3 to 10, in XLSX format and a folder with some constellation figures. I would recommend the authors add a short supplementary document indicating what information is contained in each of the files from the database, since this information cannot be directly inferred from the file names and folder structures.